# Balancing Fidelity and Diversity: Synthetic data could stand on the shoulder of the real in visual recognition

## Abstract

As generative models (GMs) advance, producing high-quality, photorealistic synthetic images is now feasible and increasingly common. Beyond use in entertainment, such synthetic data offers a promising solution to data scarcity in AI research. With the growing research interest in using synthetic data, a critical question arises: How can we maximize performance gains for downstream tasks using synthetic data? Fine-tuning or prompt engineering are two common strategies to fully exploit the potential of GMs to produce task-specific synthetic data. However, unlike these tedious optimizations, is it possible to exploit the data potential by selecting an optimal synthetic subset from a given pool? Motivated by this question, we propose an efficient data selection strategy to improve the utility of existing synthetic datasets without adjusting the GMs' output. Experiments on several benchmarks demonstrate that balancing the trade-off between fidelity and diversity in synthetic data benefits model performance and robustness. In summary, this paper presents a practical and scalable approach to harnessing synthetic data, particularly valuable in scenarios where customizing the outputs of generative models is infeasible.

## 1 Introduction

Generative models (GMs) have emerged as powerful tools for producing high-quality synthetic data and offering a promising solution for data scarcity in data-intensive AI. Recent studies demonstrate that GMs can effectively replicate datasets (*e.g.,* CIFAR-10, ImageNet), and improve recognition models trained on such synthetic data in generalization, transferability, and in-domain accuracy (Zheng et al., 2023; Sariyildiz et al., 2023; Azizi et al., 2023; He et al., 2023; Li et al., 2024; Fan et al., 2024).

However, despite their proven effectiveness, the use of synthetic data still faces practical challenges: 1) synthetic datasets may contain low-fidelity or mislabeled samples that introduce harmful noise into training; 2) they can inadvertently over-represent dominant patterns, transferring inherent biases to downstream models; 3) although expanding synthetic datasets to massive scales can alleviate these issues, the benefit comes at the cost of increased computational overhead and longer training times.

Figure 1: The diagram shows the process of data selection while considering both fidelity and diversity. Real data is first divided into the Homo-Set and the Hetero-Set. Synthetic instances are then scored by compared with these two partitions in feature space for subsequent selection.

Harnessing the potential of synthetics, prior research has primarily focused on *advancing and customizing the generation process* to produce synthetic data with higher fidelity and diversity. Fidelity is typically improved through fine-tuning (Azizi et al., 2023; Kim et al., 2024) or guidance mechanisms (Yuan et al., 2024; Kim et al., 2025) that align synthetic outputs with the target distribution, while diversity is encouraged through conditional sampling (Dunlap et al., 2023; Hemmat et al.,

2024), prompt engineering (Shipard et al., 2023) and *etc.* While effective, these approaches could be computationally expensive, require domain expertise, and are tightly coupled to specific generative models or settings, limiting their practicality.

An alternative direction is *post-generation curation*, analogous to the filtering track in Data-Comp Gadre et al. (2023), which selects a synthetic subset from a given data pool. Aligning generative data with real target data is a way to compose desired synthetic datasets, and existing methods are largely similarity-based: Image–Label Alignment methods assume that high-quality samples align with generation labels and often use pre-trained discriminators to filter out noisy instances (Chen et al., 2023c; Zhang et al., 2021b; He et al., 2023); while Image–Image Alignment prioritizes synthetic samples that are highly similar to real images to mimic the target distribution (Kynkäänniemi et al., 2019; Lin et al., 2023). These strategies emphasize fidelity, but largely neglect diversity, reducing the utility of synthetic data sets on a scale (Fan et al., 2024) due to repeated patterns and missing novel information.

In this work, we propose a *fidelity–diversity* balanced *post-generation* curation framework for selecting synthetic data from static data pools. We begin by empirically analyzing the fidelity–diversity trade-off in real data, showing that downstream performance depends on effectively managing this balance. Guided by this insight, we propose to partition the real dataset into two subsets: a **Homogeneous** (HOMO) set characterized by high internal similarity, and a **Heterogeneous** (HETERO) set enriched with variation. Building on this partitioning, we develop a data selection framework that scores and selects synthetic instances to balance high semantic quality with representational diversity (see Fig. 1). Specifically, we compute two complementary metrics: (1) *fidelity scores*, measuring semantic similarity to real samples, and (2) *diversity scores*, quantifying deviation from repetitive patterns in the HOMO. By combining these metrics, our method prioritizes synthetic instances that are both semantically faithful and diverse.

To quantify how synthetic dataset selection affects downstream performance, we conduct extensive experiments across datasets (*e.g.,* CIFAR-10, Tiny-ImageNet, ImageNet) and downstream models (*e.g.,* ResNet, ViT). We first synthesize diverse candidate pools using multiple generators trained on target datasets and then apply a post-generation curation process to select an optimal training set. Models trained on curated data (from scratch or fine-tuning) are then benchmarked for in-domain accuracy and out-of-domain (OOD) generalization. The results show that our strategy outperforms prior data selection approaches in both evaluation settings. These results underscore that the post-generation processing of synthetic data is as crucial as the generation itself. In summary, our contributions are as follows:

1. We introduce a data-splitting strategy that partitions real data into Homogeneous (HOMO) and Heterogeneous (HETERO) subsets, enabling fidelity–diversity balanced selection.

2. We propose a principled, post-generation selection strategy that jointly quantifies and balances fidelity and diversity. The framework is generator-agnostic, requiring only a synthetic data pool, and avoids costly retraining or fine-tuning of generators.

3. We validate our method across diverse generators, datasets, and recognition models, showing consistent improvement over existing curation strategies in both in-domain accuracy and out-of-domain generalization.

## 2 RELATED WORKS

### 2.1 GENERATIVE DATA IN MODEL TRAINING

Realistic synthetic data generated by GMs present a promising resolution for data scarcity, motivating growing research into their impacts in AI community. For example, Hammoud et al. (2024); Tian et al. (2023a;b) train models entirely on synthetic data to learn robust visual representations. Zhou et al. (2023); Azizi et al. (2023); He et al. (2023) demonstrate the effectiveness of synthetic data for training image recognition models. In addition, Qraitem et al. (2024); Zeng et al. (2025); Sariyildiz et al. (2023) show that the combination of real with synthetic data further enhances the performance of the model. On the other hand, the ubiquity of synthetic data also raises concerns about its latent limitations. Hataya et al. (2023); Fan et al. (2024) demonstrate that the distribution gap between real and synthetic data can negatively impact model performance. Similarly, Li et al.

(2024); Wang et al. (2024) emphasize that realizing the full benefits of synthetic data requires careful tailoring in contrastive and transfer learning settings. As GMs evolve rapidly, the influence of synthetic data on the AI community is undeniable, as demonstrated by existing research. Our work goes beyond these advances by establishing data selection as a critical mechanism that substantially improves both the effectiveness and efficiency of synthetic data for downstream tasks.

## 2.2 Synthetic Data Curations

**Fidelity-Guided Curation.** High-fidelity generations ensure semantic correctness, making fidelity-based selection effective for curating quality datasets. Such curation strategies can be broadly categorized into two main approaches: **1) Image-Label Alignment:** Bhattarai et al. (2020); Zhang et al. (2021a); Chen et al. (2023b); Vu et al. (2021) use pretrained or task-specific models to filter low-quality synthetic samples by discarding those misclassified within top-$k$ predictions. Recently, He et al. (2023); Lin et al. (2023); Tang et al. (2025) use CLIP to directly assess image-label semantic alignment in the embedding space; **2) Image-Image Alignment:** Kynkäänniemi et al. (2019); Naeem et al. (2020); Chen et al. (2023a); Lin et al. (2023) measure the similarity between synthetic and real images, discarding those far from the real distribution. Lin et al. (2023) cluster real data and using cluster centroids as anchors to retrieve synthetic samples. However, both these two approaches face inherent limitations. The former depends heavily on the performance of pretrained models, while the latter may compromise diversity by focusing too much on visual similarity.

**Diversity-Guided Curations.** As generative images become increasingly realistic, diversity is key for the utility of synthetic datasets. Prior works enhance diversity by varying prompts (Shipard et al., 2023), applying text-conditioned augmentation (Dunlap et al., 2023; da Costa et al., 2023), using textual inversion (Trabucco et al., 2024), or conditioning on classifier outputs (Hemmat et al., 2024). While effective, these methods require fine-tuning or prompt engineering and focus on generation-time diversity, offering little guidance on how to efficiently leverage existing synthetic datasets.

Building the priors, we curate informative synthetic datasets by jointly considering fidelity and diversity, without modifying generative model output, forming a post-generation data curation pipeline.

## 3 Methods

Training in generative data, achieving an optimal balance between fidelity and diversity is vital. Synthetic data that closely resemble real data offers limited novelty, while excessive diversity introduces noise, negatively impacting model performance. This section outlines our approach to manage this trade-off. We begin with a preliminary analysis quantifying the trade-off between fidelity and model performance, then introduce a framework to categorize data based on its characteristics. Finally, we propose a selection strategy that leverages both fidelity and diversity.

### 3.1 Preliminary Analysis: Measuring the Impact of Similarity and Diversity

We first conduct a preliminary analysis to quantify the relationship between fidelity and model performance. Given a dataset $D$, we apply traditional augmentation techniques—including random cropping, rotation, and color jittering—to control fidelity levels. Specifically, we test random cropping with padding sizes of 1 ($C1$), 2 ($C2$), 4 ($C4$), and 8 ($C8$). We then incorporate rotations of 5°($+R5$), 10°($+R10$), and 15°($+R15$), followed by color jittering intensities of 0.2 ($+CJ0.2$) and 0.4 ($+CJ0.4$). As augmentation strength increases, samples become less similar to the original data. To quantify this relationship, we measure image similarity using the Fréchet Inception Distance (FID) and text-image alignment using the CLIP score. Fig. 2 demonstrated that as augmentation

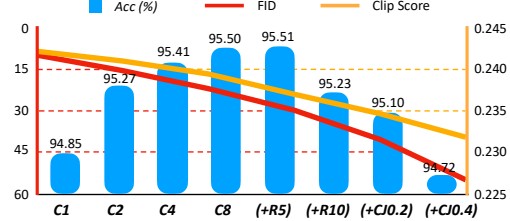

Figure 2: Impact of augmentation strength on fidelity and model performance in CIFAR10. As strength increases ($C1 \rightarrow ... +R5 ... \rightarrow +CJ0.4$), dataset similarity (FID, CLIP score) decreases, indicating reduced resemblance to the original. Models' performance follows a non-monotonic trend, improving at moderate augmentation levels before declining at higher levels.

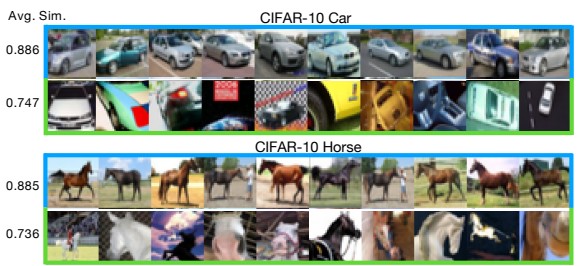

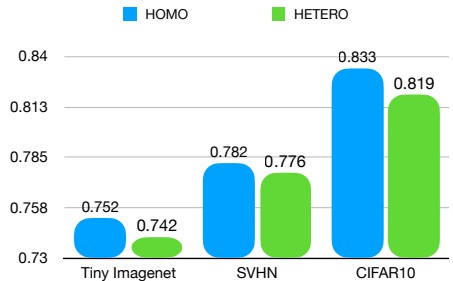

Figure 3: HOMO and HETERO instances in the CIFAR-10. *Avg. Sim.* represents the average similarity between the images in each row and the entire class. HOMO instances are more representative and better express the core semantics. HETERO instances are more diverse, capturing a broader range of variations (More examples are in the Appendix Fig.11).

Figure 4: Similarity between the synthetic and HOMO, HETERO splits. We measure cosine similarity between HOMO/HETERO subsets and generations of models trained by each real dataset. Synthetic data consistently more closely resembles HOMO than HETERO.

strength increases, both similarity metrics (*i.e.,* FID, CLIP score) decrease. However, model performance initially improves before declining beyond a certain augmentation threshold. We observe a non-monotonic relationship between fidelity and downstream model performance: moderate augmentation improves performance, but excessive diversity introduces noise, leading to degradation. This finding highlights the importance of a balanced selection strategy that optimally incorporates both fidelity and diversity.

While this analysis highlights the trade-off, navigating it effectively requires a structured approach. To achieve this, we introduce a framework for categorizing synthetic samples based on their similarity characteristics, allowing for a more principled selection of training data.

## 3.2 CATEGORIZING HOMOGENEOUS AND HETEROGENEOUS SAMPLES

To guide the selection of synthetic data, we first categorize real images within a given class into two distinct sets: **Homogeneous (HOMO)** and **Heterogeneous (HETERO)**. HOMO instances represent the canonical semantics, exhibiting high intra-class similarity in feature space. HETERO instances capture greater variation, including less typical instances that contribute to diversity.

**Identifying HOMO and HETERO Instances.** We introduce a strategy to systematically identify HOMO and HETERO instances within each class. We formally define these categories by first extracting image features using a pretrained model (*e.g.,* MoCo v3 (Chen* et al., 2021)). Given an image set $\mathcal{I} = \{I_1, I_2, \ldots, I_n\}$, we obtain the corresponding feature set $\mathcal{F} = \{f_1, f_2, \ldots, f_n\}$.

To quantify feature similarity, we compute the cosine similarity between each pair of features:

$$\cos(f_i, f_j) = \frac{f_i \cdot f_j}{|f_i||f_j|}, \quad \forall i \neq j \tag{1}$$

For each feature $f_i$, we identify its most similar counterpart $f_{j^*(i)}$ within the same class:

$$j^*(i) = \arg\max_{j \neq i} S(f_i, f_j) \tag{2}$$

To classify samples, we define the HOMO set as the group of instances that serve as the most similar counterpart to at least one other instance. We iterate over all features and collect those have been retrieved as the closest match at least once within the class:

$$\mathcal{I}_{\mathcal{HO}} = \{I_i | \exists j \neq i, j^*(j) = i\} \tag{3}$$

Conversely, HETERO set is defined as the complement of HOMO. These instances are not the most similar to any other feature in the dataset, representing unique variations within the class:

$$\mathcal{I}_{\mathcal{HE}} = \mathcal{I} \setminus \mathcal{I}_{\mathcal{HO}} \tag{4}$$

**HOMO –HETERO Analysis and Impact.** To better understand the characteristics of the HOMO and HETERO sets, we present a toy example in CIFAR-10. In Fig. 3, HOMO samples exhibit higher intra-class similarity than HETERO samples both visually and quantitatively.

Upon this, experiments further demonstrate that ***GMs more readily learn and reproduce* HOMO *set containing repeated canonical pattern***. In Fig. 4, quantifying how synthetic data aligns with the HOMO and HETERO splits, the results show the synthetic data closely resembles the HOMO sets. GMs exhibit an inherent preference for HOMO set, despite the comparable volumes of the two sets. This bias may also help explain the diversity limitations observed in synthetic datasets.

Beyond the impact in GMs, in real dataset, ***discriminators also achieve higher performance on the* HOMO *set while struggling with its* HETERO *counterpart***. This suggests that, despite sharing the same underlying semantics, HETERO contains inherently more challenging real instances, which the discriminator remains weak at modeling (extra experiments in Fig. 12 in the Appendix).

Combining these observations, using synthetic datasets without accounting for the characteristics of real data in HOMO and HETERO can be risky. Since GMs tend to produce HOMO-like samples, models trained on such data may suffer reduced robustness and even an amplified gap between canonical and non-canonical cases. Therefore, in the following section, we present a synthetic data selection strategy that addresses these issues, which stem from the real dataset itself.

### 3.3 SYNTHETIC DATA SELECTION STRATEGY

After getting HOMO and HETERO sets, we propose a novel synthetic data selection strategy to identify desirable instances from given pools. The core objective is to ensure high semantic fidelity while enhancing diversity by prioritizing samples that diverge from the dominant patterns in HOMO.

**Partitioned Selection Strategy.** Unlike prior methods that treat the entire real dataset as a single reference for data selection, we propose a partitioned selection approach. We treat HOMO and HETERO as separate partitions and select synthetic instances by referring to real images within

each partition under the same principle: desired synthetics should be sufficiently close to their corresponding real instances, while prioritizing those that deviate from canonical representations. By combining the selection results from HOMO and HETERO, we obtain the final selected pool.

Concretely, We compute a selection score $S^p$ for each synthetic sample based on its correlation with partition

---

**Algorithm 1** Data Selection with HOMO and HETERO Partitions

**Require:** Feature maps $\mathcal{F}^{Homo} \in \mathbb{R}^{n \times d}$, $\mathcal{F}^{Heter} \in \mathbb{R}^{m \times d}$, $\mathcal{F}^{syn} \in \mathbb{R}^{s \times d}$; centroid $\mathcal{C}^{Homo} \in \mathbb{R}^{d}$; hyperparameter $\alpha \in [0, 1]$

**Ensure:** Selected indices $\mathcal{I}_{selected}$

1: $\text{idx} \leftarrow \arg\max(\mathcal{F}^{Heter} \cdot \mathcal{F}^{Homo\top}, \text{axis} = 1)$
2: $\mathcal{F}^{ref} \leftarrow \mathcal{F}^{\text{HOMO}}[\text{idx}]$
3: **for** $p \in \{Homo, Hetero\}$ **do**
4:      Fidelity: $S^p_{fid} \leftarrow \cos(\mathcal{F}^{syn}, \mathcal{F}^p)$
5:      Reference: $\mathcal{R}^{ref} \leftarrow \begin{cases} \mathcal{C}^{Homo}, & p = Homo \\ \mathcal{F}^{ref}, & p = Heter \end{cases}$
6:      Diversity: $S^p_{div} \leftarrow -\cos(\mathcal{R}^{ref} - \mathcal{F}^p, \ \mathcal{F}^{syn} - \mathcal{F}^p)$
7:      Score: $S^p \leftarrow \alpha S^p_{div} + (1 - \alpha) S^p_{fid}$
8:      $\mathcal{I}^p_{\text{top}} \leftarrow \text{Top}_k(S^p)$
9: **end for**
10: **return** $\mathcal{I}_{\text{selected}} \leftarrow \mathcal{I}^{\text{HOMO}}_{\text{top}} \cup \mathcal{I}^{\text{HETERO}}_{\text{top}}$

---

$p \in \{\text{HOMO}, \text{HETERO}\}$. Samples are then ranked within each partition, and the instances with high $S^p$ get retrieved. See Alg. 1 for the pseudo-code of our selection procedure.

**Scoring Mechanism for Each Partition.** Our selection mechanism balances two competing objectives: the *fidelity score*, which measures how closely a synthetic instance resembles real data, and the *diversity score*, which measures how much it deviates from canonical patterns in the real dataset.

*1) Fidelity Score* $S^p_{\text{fidelity}}$ quantifies how well a synthetic sample aligns with the real distribution of partition $p$. We measure it using the cosine similarity between each pair of features in the synthetic set $\mathcal{F}^{syn}$ and the real set $\mathcal{F}^p$, as defined in Eq. 5. Higher fidelity scores indicate that the synthetic samples closely resemble real data in given partitions.

$$S^p_{fidelity} = \cos\left(\mathcal{F}^{syn}, \mathcal{F}^p\right), \tag{5}$$

*2) Diversity Score* $S^p_{\text{diversity}}$ is designed to encourage the selection of synthetic samples that deviate from canonical patterns. Specifically, in HOMO, diversity is quantified as the deviation of synthetic samples from the centroid of HOMO. In HETERO, diversity is measured as the deviation between the vector from a real image in HETERO to synthetic data and the vector from the same real image

to its most similar match in HOMO (extra visualization in Fig.13 of the Appendix). The diversity metric is defined in Eq. 6, where $\mathcal{R}^{ref}$ denotes the reference features in real dataset.

$$S^p_{diversity} = -\cos(\mathcal{R}^{ref} - \mathcal{F}^p, \ \mathcal{F}^{syn} - \mathcal{F}^p), \tag{6}$$

As illustrated in Fig. 5, if these two vectors point in opposite directions, the cosine similarity is negative, leading to a higher diversity score. If the vectors are aligned, the synthetic sample is closer to the homogeneous distribution, resulting in a lower diversity score.

*3) Total Score.* To control the trade-off between fidelity and diversity, we define the score as:

$$S^p = \alpha \cdot S^p_{\text{diversity}} + (1 - \alpha) \cdot S^p_{\text{fidelity}}, \tag{7}$$

where $\alpha$ is a hyperparameter that determines the selection priority. $\alpha = 0$ (*MaxSim*) will prioritize fidelity, ensuring that synthetic instances closely resemble real ones. $\alpha = 1$ (*MaxDiv*) will prioritize diversity, encouraging synthetic samples that deviate from dominant real data patterns. Adjusting $\alpha$, we can dynamically control the selection behavior based on the specific needs of the application. Obtain $S^p$, where a higher value indicates a higher quality for the corresponding synthetic instance.

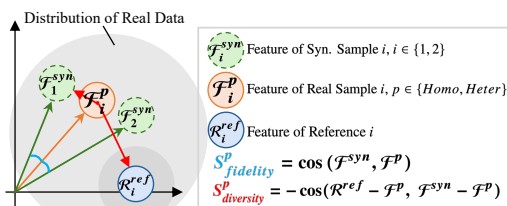

Figure 5: Illustration of the computation of $S^p_{\text{fidelity}}$ and $S^p_{\text{diversity}}$ in partition $p$. Fidelity is assessed as $\cos\left(\mathcal{F}^{\text{syn}}_i, \mathcal{F}^p_i\right)$, while diversity is measured by the angle between vectors ending at $\mathcal{F}^{\text{syn}}_i$ and $\mathcal{R}^{ref}_i$. In the diagram, Syn. Sample 1 exhibits greater diversity than Syn. Sample 2.

## 4 EXPERIMENT AND RESULTS

In experiment, we first use GMs to build synthetic data pools and curate datasets via different curation strategies. To assess their impact, we evaluate downstream models trained on these datasets in both in- and out-of-domain settings. Concretely, we adopt GMs and classifiers from public repositories (summarized in Appendix Tab. 5), and set $\alpha = 0.5$ in Eq. 7 throughout the experiments.

### 4.1 DATASETS AND BASELINES

**Datasets.** Unlike works focusing on improving generation quality, we aim to identify informative samples from a given synthetic dataset. Therefore, we employ the off-the-shelf EDM (Wang et al., 2023) to synthesize **SVHN** (Netzer et al., 2011), **CIFAR-10** (Krizhevsky et al., 2010), and **Tiny ImageNet** (Le & Yang, 2015), on which the models are pretrained, enabling evaluation across datasets of increasing complexity (10, 10, and 200 classes, respectively). For further validating, we test on **ImageNet-1K** (IN-1K) generated by EDM2 (Karras et al., 2024b;a) confirming the scalability of our strategy. To comprehensively evaluate the impact of the curated synthetic datasets, we perform in-domain validation on 4 test sets corresponding to each training dataset, and out-of-domain validation on 9 derivative datasets, as discussed in Sec. 4.3. Totally, 13 datasets are used in our experiments.

**Baselines.** To benchmark our method, we compare it against 3 baseline categories: 1) *naïve sampling*, represented by **RandSelect** (Wood et al., 2021; Carlini et al., 2023; Kim et al., 2023), an uninformed random-selection strategy; 2) *image–label alignment (I–L–Align)*, embodied by **CLIP-Align** (He et al., 2023), which leverages CLIP embeddings to match synthetic images with their textual labels; and 3) *image–image alignment (I–I–Align)*, exemplified by **RealScore** (Kynkäänniemi et al., 2019), quantifying the realism of synthetic images, and **SBSim** (Lin et al., 2023), which selects samples according to their similarity to real images (equivalent to setting $\alpha = 0$ in Eq. 7).

### 4.2 SCALING MODEL PERFORMANCE WITH SELECTIVE SYNTHETIC DATA

Demonstrating the gain of data selection, we train models with synthetic samples and evaluate them on real test data. We apply various selection strategies and train models, *e.g.,* EfficientNet-B0 (Tan & Le, 2019), ResNet-18/50 (He et al., 2016) and ViT (Wu et al., 2020) to quantify the gains. To simulate datasets with various complexity, experiments are conducted on SVHN, CIFAR-10, and Tiny-ImageNet. For large-scale validation, we further evaluate on IN-1K. The experiment results highlight the importance of curating synthetic datasets to maximize downstream model performance.

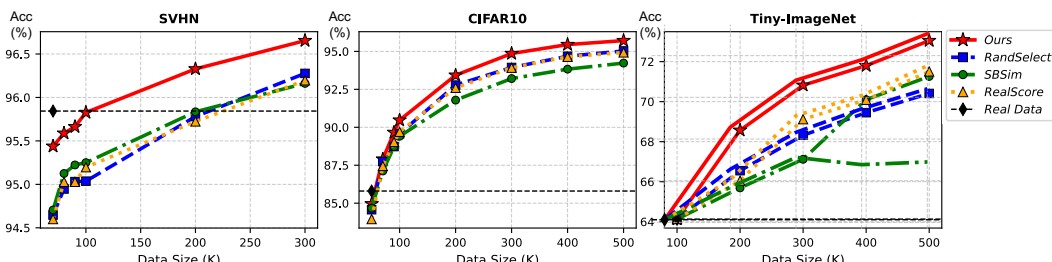

Figure 6: Average accuracy over **6 runs** when trained on different synthetic datasets. We compare our method with *RandSelect*, *SBSim*, *RealScore* (*CLIP-Align* is missing in the figure for the poor performance, and detailed results are in appendix Tab.6, 7, 8 ). As synthetic data volume grows, accuracy surpasses the real-data baseline (black dashed line with diamond marker) across datasets. Our approach consistently selects higher-quality samples, yielding superior downstream performance.

In **SVHN**, Fig. 6 (left) shows the performance of ResNet-18. Our strategy attains comparable accuracy to a real-data baseline using 100K synthetic samples, whereas alternative methods require over 200 k samples to match this performance. In **CIFAR-10**, selecting from online synthetic dataset (5M), (Fig. 6, middle), ResNet-18 requires a comparable size of data to reach the real-data baseline across all strategies. However, as volume increases, our strategy gains a clear advantage: it achieves 95% accuracy with 300 k samples, while *RandSelect* and *RealScore* each require around 500 k, and *SBSim* demands substantially more. We attribute this divergence to enhanced data diversity: beyond a certain volume, redundant samples offer little information gain, while our selections continue to enrich the training set. On **Tiny-ImageNet**, we leverage synthetic data as a form of real-data augmentation to mitigate the negative effects of domain shift in this complex setting. As shown in Fig. 6 (right), our strategy allows ResNet-50 to achieve higher accuracy with fewer data. To further assess the impact of similarity- and diversity-only selection, we show that by jointly accounting for both fidelity and diversity, our method achieves the best performance, underscoring the importance of curating datasets to balance these two factors (see Tab. 8 in the Appendix).

Table 1: ImageNet-1K Top-1 Acc. (%) with 1M and 3M selected samples. Background colors indicate different models: ViT-B/16 and ResNet-50 , evaluated under both **from-scratch** and **fine-tuning** settings. The corresponding training recipes are provided in Appendix A.5.1.

| Selection | Scratch | | | | | | Fine-tune | | | |
|---|---|---|---|---|---|---|---|---|---|---|
| | +0 | +1M | +3M | +0 | +1M | +3M | 1M | 3M | 1M | 3M |
| Random | 64.22 | 70.90 | 73.54 | 69.27 | 72.03 | 72.13 | 78.52 | 78.68 | 71.90 | 73.65 |
| Realism | - | 70.92 | 73.53 | - | 72.01 | 72.16 | 78.52 | 78.67 | 71.93 | 73.62 |
| CLIP-Align | - | 69.18 | 70.88 | - | 69.44 | 70.98 | 72.40 | 75.23 | 62.57 | 67.57 |
| SBSim ($\alpha = 0$) | - | 70.95 | 73.23 | - | 71.97 | 72.18 | 78.58 | 78.76 | 72.18 | 74.14 |
| Diversity ($\alpha = 1$) | - | 71.42 | 73.79 | - | 71.80 | 73.25 | 78.72 | 79.14 | 72.25 | 74.25 |
| Ours | - | **71.49** | **74.02** | - | **72.09** | **73.33** | **78.97** | **79.36** | **73.14** | **74.76** |

Evaluation is conducted on **IN-1K** for a large-scale setting. Prior works (Sariyildiz et al., 2023; Fan et al., 2024; Azizi et al., 2023) have shown that current GMs fail to capture the full modes of IN-1K leading to performance gap. Thus, more practical strategies on the ImageNet scale involve 1) training models from scratch with synthetic data as augmentation and 2) fine-tuning pre-trained models using synthetic data alone. We first generate 10M synthetic images using EDM2, and then train ViT and ResNet-50 with different scales of selected synthetic subsets. The experimental results are reported in Tab. 1. Echoing the previous findings, the experimental results further highlight the importance of balancing fidelity and diversity in synthetic datasets.

### 4.3 ENHANCING MODEL GENERALIZABILITY WITH SELECTED SAMPLES

Models trained on informative and diverse data are expected to achieve greater robustness under Out-of-Distribution (OOD) conditions. In this part, we evaluate the generalizability of the best models trained (from scratch) in the previous section with OOD datasets. The results demonstrate that curated synthetic data improves model generalizability, validating our selection strategy.

**OOD of Tiny-ImageNet.** We utilize Tiny-ImageNet-C (Hendrycks & Dietterich, 2019a) for the evaluation of OOD, which incorporates various types of corruption. We classify them into three types: color-variation set (*i.e.,* brightness adjustment, contrast variation), noise-variation set (*i.e.,* pixelation, Gaussian noise, motion blur), and compression-variation set (*i.e.,* JPEG compression). The results are presented in Fig. 7(a).

**OOD of CIFAR-10.** We use CIFAR-10-Warehouse (Sun et al., 2024) as a benchmark, which is collected from diverse sources like search engines (*e.g.,* Flickr (2025), Pexels (2025)) and diffusion models. The results are presented in Fig. 7(b).

**OOD of IN-1K.** We use several OOD datasets and categorize them into two groups 1) *original suite*, including ImageNet-V2 (Recht et al., 2019) and ImageNet-Sketch (Wang et al., 2019); and 2) *derivative suite*, including ImageNet-C (Hendrycks & Dietterich, 2019b), -Drawing, and -Cartoon (Salvador & Oberman, 2022), derived from the validation set. See Appendix A.3.2 for details. The evaluation results is summarized in Fig. 7(c),(d).

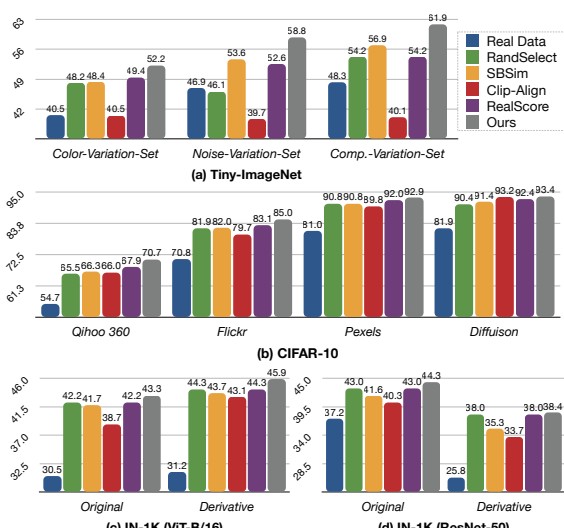

Figure 7: OOD Evaluation. Among different settings, ours consistently achieves the best performance.

**OOD of SVHN.** We evaluate OOD performance using the extra split of SVHN (Netzer et al., 2011) and test subset of distorted SVHN. Detailed results are presented in Tab. 9 in the Appendix.

## 5 ANALYSIS AND DISCUSSION

**HOMO and HETERO, the Characteristics of Real Data Impacts.** When splitting HOMO and HETERO within a class across datasets, the number of samples in each subset remains nearly equal, suggesting that *approximately half of the samples effectively capture the primary semantics of the class.*

Within individual dataset, when selecting synthetic data using HOMO and HETERO as references, we observed that although the two sets have comparable volumes, HOMO yields fewer unique synthetic instances. We attribute this to the *higher intra-class similarity within HOMO (Tab. 3 in the Appendix), which makes it more challenging for GMs to capture fine-grained variations among semantically similar concepts, compared to the more diverse HETERO setting.*

Across datasets, under the same setting, we find that the number of unique synthetic samples selected is proportional to dataset complexity. Specifically, the number of unique instances follows the order: IN-1K > Tiny-ImageNet > CIFAR-10 > SVHN. This observation suggests that *selecting diverse synthetic samples is inherently more difficult in simpler datasets with lower intra-class variation.*

Table 2: Cosine similarity between High-CLIP-Score samples and HOMO/HETERO subsets across encoders in CIFAR-10.

| Subset | CLIP | MoCo | ConvNeXt |
|--------|------|------|----------|
| HOMO | 0.946 | 0.934 | 0.839 |
| HETERO | 0.943 | 0.914 | 0.805 |

**The Impact of Diversity and Fidelity.** Fidelity and diversity are two factors to measure quality of synthetic data. In Fig. 6, integrating the diversity factor, our method achieves the best performance as dataset size increases. Comparing ours with ***SBSim*** which does not consider diversity, in SVHN and CIFAR-10, ***SBSim*** underperforms at larger scales, indicating that fidelity-only methods become less informative as dataset size increases. More critically, its performance falls below random selection, indicating that fidelity alone is insufficient and that overemphasis on it can even be detrimental. On Tiny-ImageNet, ***SBSim*** outperforms *RandSelect* at 500K samples, suggesting that the benefits of fidelity have not yet saturated in this setting. High-fidelity selection still enhances training by better preserving primary semantics. *The findings suggest that the validity of fidelity-only methods*

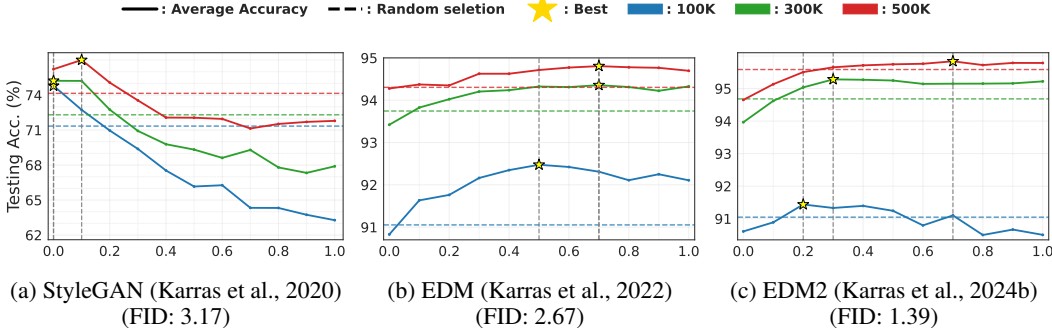

(a) StyleGAN (Karras et al., 2020) (FID: 3.17)

(b) EDM (Karras et al., 2022) (FID: 2.67)

(c) EDM2 (Karras et al., 2024b) (FID: 1.39)

Figure 8: Testing accuracy versus trade-off $\alpha$ (0 = fidelity, 1 = diversity). Each subfigure compares the 100K, 300K, and 500K settings with trend lines, and the optimal point on each curve is marked by a star. Results are reported with **ResNet-18** using the **average accuracy over 8 runs**.

*is influenced by dataset volume and characteristics, with diversity emerging as a critical factor, particularly for current advanced GMs.*

Besides **SBSim**, the **CLIP-Align** also reveals a critical issue: the overlooking of diversity. As shown in Tab. 1, and appendix Tab. 6, 7, 8, **CLIP-Align** gets the worst performance. This is primarily due to that *filters like CLIP favors canonical patterns, preserving semantic fidelity but at the cost of diversity.* Consequently, prioritizing high CLIP-score images produces a more monotonous dataset as the scale increases (Tab.2, Fig.16, 17 in Appendix A.4.1).

**Impact of Generators and Data Scales on the Optimal Fidelity-Diversity Balance ($\alpha$).** We explore the relationship between $\alpha$ in Eq. 7 and different GMs. Our experiments use 1M synthetic CIFAR-10 samples from three GMs. As shown in Fig. 8, the optimal $\alpha$ depends on both the quality of the synthetic data and the volume of training data. (1) For StyleGAN2 (Fig. 8a), which yields the highest FID, fidelity remains the key factor for classification performance; (2) When the generated distributions are closer to the real (as in EDM and EDM2), Figs.8b, 8c demonstrate the benefit of tuning the trade-off factor $\alpha$. (3) Arcoss all the datasets, as the training volume increases, the optimal $\alpha$ shifts toward larger values, underscoring the increasing importance of diversity at larger scales, where canonical high-fidelity samples provide limited additional gains. (4) In terms of the best $\alpha$ in each GMs, the biggest shift is observed in the most advanced EDM2, suggesting that with larger training volumes and highly realistic generations, diversity becomes the decisive factor.

**Impact of Feature Extractors.** Since our strategy relies on image features, we examine whether the choice of feature extractor matters. We conduct an ablation study using three representative encoders, each reflecting a distinct training paradigm: SigLIP (text–image alignment) (Zhai et al., 2023), DINOv3 (self-supervised) (Siméoni et al., 2025), and ViT (supervised). Evaluations are performed on two datasets with different image resolutions, CIFAR-10 (32×32) and ImageNet-100 (224×224) (detailed configurations are summarized in Appendix A.5). The results, summarized in Fig. 9, show that

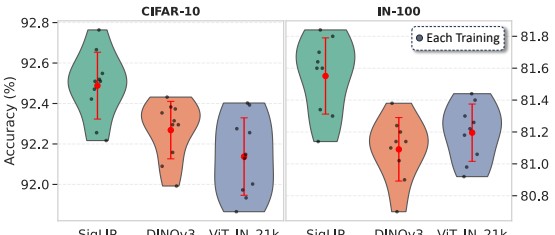

Figure 9: Testing accuracy of models trained on selected synthetic data with different feature extractors. The corresponding training configurations are provided in Appendix A.5.

although SigLIP achieves the best performance, the absolute differences among extractors remain modest. This suggests that SigLIP is a promising choice, but its impact is relatively minor compared to the quality of synthetic data and the choice of $\alpha$ in the proposed strategy.

## 6 CONCLUSIONS

We propose a data selection method that enhances the utility of synthetic datasets without altering the generative process. By balancing fidelity and diversity with a novel HOMO-HETERO splitting strategy, our approach improves model accuracy and generalizability, demonstrating the importance of strategic selection in leveraging synthetic data. Our findings indicate that optimized selection enables comparable performance with fewer synthetic samples. As synthetic data quality improves, our work underscores the increasing significance of selection beyond mere similarity.

## 7 ETHICS STATEMENT

Our work on synthetic data selection raises several ethical considerations that warrant discussion. **Bias Amplification**: Our HOMO-HETERO partitioning strategy could potentially amplify biases present in the original data if the HOMO set disproportionately represents certain demographic groups or patterns. We mitigate this by explicitly incorporating diversity metrics in our selection criteria. **Environmental Impact**: While synthetic data generation incurs computational costs, our selective curation approach reduces the environmental footprint by enabling comparable performance with smaller datasets, requiring fewer training iterations and less storage. **Privacy Preservation**: Although synthetic data generally poses fewer privacy risks than real data, our method does not specifically filter for instances that might inadvertently resemble real individuals. Practitioners should implement additional privacy checks when deploying our method in sensitive domains. **Responsible Use**: We acknowledge that synthetic data technology could be misused for creating deceptive content. Our method is designed for legitimate research and educational purposes, specifically to improve model training efficiency while maintaining performance standards. We encourage adherence to established guidelines for synthetic data use and advocate for continued development of detection methods to distinguish synthetic from real content.

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

# A APPENDIX

## A.1 CATEGORIZING HOMOGENEOUS AND HETEROGENEOUS SAMPLES

### A.1.1 CHARACTERISTICS OF HOMO AND HETERO DATASET

In real datasets, we introduce a novel split into two subsets: HOMO, containing canonical instances, and HETERO, comprising instances with greater intra-class diversity. Fig. 11 illustrates the characteristics of these subsets both visually and quantitatively. The HOMO set exhibits higher average cosine similarity among image pairs, whereas the HETERO set is more diverse. Extending to more datasets, as shown in Tab. 3, this pattern remains consistent across them.

Table 3: Average intra-class similarity of HOMO and HETERO subsets across datasets. HOMO consistently gets higher intra-class similarity.

| Subset | SVHN | CIFAR-10 | Tiny-ImageNet | ImageNet-1K |
|---|---|---|---|---|
| HOMO | 0.8507 | 0.7948 | 0.8236 | 0.8000 |
| HETERO | 0.8385 | 0.7887 | 0.7894 | 0.7725 |

### A.1.2 IMPACT OF HOMO AND HETERO FOR DOWNSTREAM MODELS

HOMO and HETERO together constitute a complete real dataset, and when such data are used to train downstream models, these characteristics may be implicitly inherited.

For generative models, we investigate the relationship between generated instances and real instances in HOMO and HETERO. As shown in Fig.4, samples generated by EDMKarras et al. (2022) exhibit higher similarity to those in the HOMO set, highlighting the well-known limitation of reduced diversity in generative data. Since HOMO samples capture canonical patterns that are easier for generative models to learn, this bias toward HOMO leads to higher intra-class similarity compared to real datasets, thereby reducing overall diversity (as show in Fig. 10).

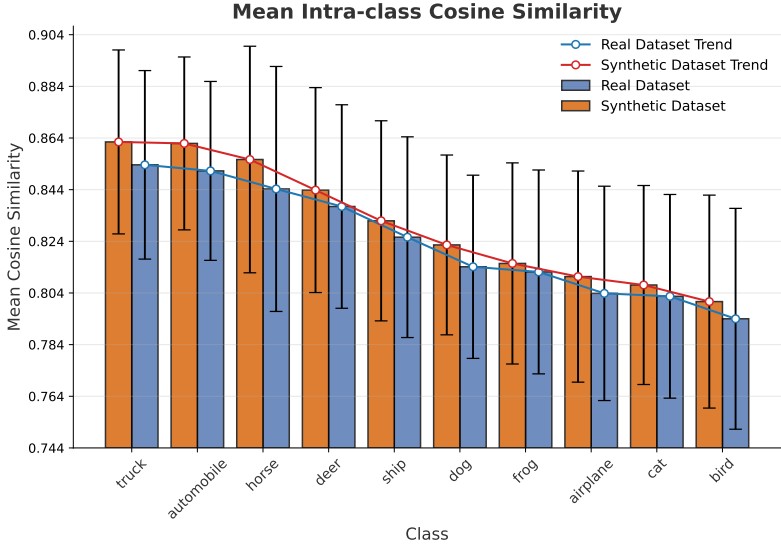

Figure 10: The cosine similarity measures how alike instances are within the individual class in CIFAR-10. From the figure, the mean cosine similarity of each class in the synthetic dataset follows the trend of the real dataset, indicating that the generative model effectively captures patterns from real data. However, the synthetic data consistently exhibit higher mean cosine similarity than the real data, suggesting that within each class, synthetic instances are more homogeneous and share greater similarity with each other compared to real instances. **This figure release that synthetic dataset have limitation of diversity in each individual class**.

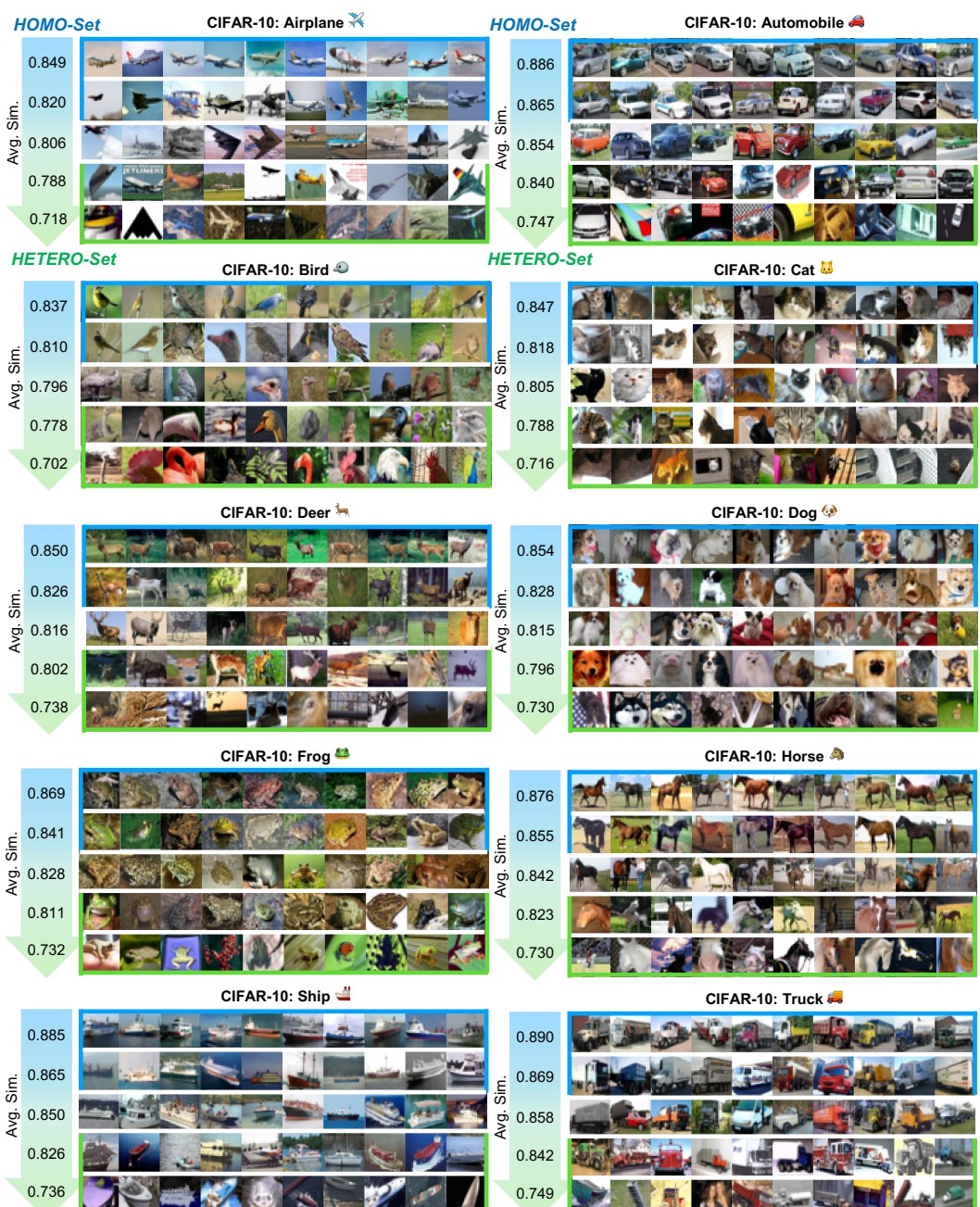

Figure 11: In the CIFAR-10 dataset, we select both HOMO and HETERO images within a specific class. HOMO instances (framed in blue) are more representative and better express the core semantics of the given class. HETERO instances (framed in green) are more diverse, capturing a broader range of variations

For discriminators trained on real data, we observe that even when evaluated on the original training set, performance on HOMO and HETERO still shows a clear gap. As shown in In Fig. 12, we use online pretrained models to perform inference on the training dataset, and observe consistently higher performance on HOMO than on HETERO. We attribute this to the presence of canonical patterns in different subsets: HOMO contains more canonical instances that are easier for models to learn, whereas HETERO requires capturing more diverse variations.

Based on this observation, training discriminators with generative data can be risky, as it may amplify the limitations already present in those trained on real data, further biasing the model toward canonical samples and degrading performance on real data inference.

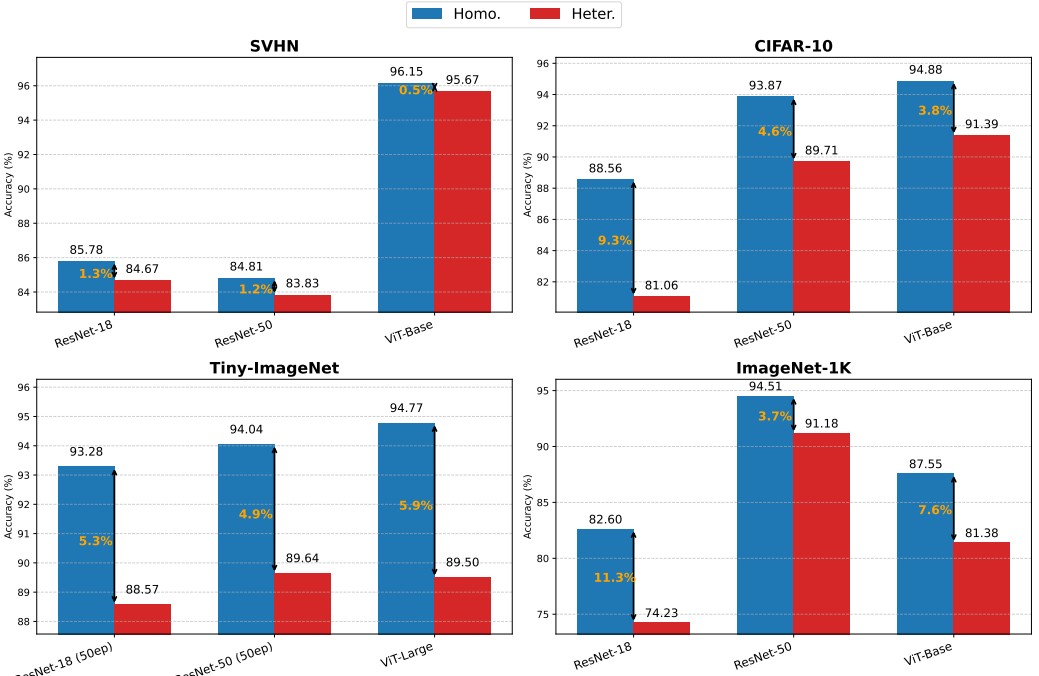

Figure 12: Performance gap between HOMO and HETERO across various datasets and pretrained models. For **SVHN**, **CIFAR-10**, **Tiny-ImageNet**, and **ImageNet-1K**, we evaluate multiple ResNet and ViT variants. Detailed descriptions and download links for all pretrained models are provided in Table 4. The experimental results reveal a striking fact: even within the training set, a clear performance gap persists between HOMO and HETERO.

Table 4: Pretrained discriminators used in our experiments.

| Dataset | Model | Link |
|---|---|---|
| SVHN | ResNet-18 | https://huggingface.co/edadaltocg/resnet18_svhn |
| | ResNet-50 | https://huggingface.co/edadaltocg/resnet50_svhn |
| | ViT-Base | https://huggingface.co/edadaltocg/vit_base_patch16_224_in21k_ft_svhn |
| CIFAR-10 | ResNet-18 | https://huggingface.co/SamAdamDay/resnet18_cifar10 |
| | ResNet-50 | https://huggingface.co/anonauthors/cifar10-timm-resnet50 |
| | ViT-Base | https://huggingface.co/nateraw/vit-base-patch16-224-cifar10 |
| Tiny-ImageNet | ResNet-18 | https://github.com/zeyuanyin/tiny-imagenet |
| | ResNet-50 | https://github.com/zeyuanyin/tiny-imagenet |
| | ViT-L | https://github.com/ehuynh1106/TinyImageNet-Transformers |
| ImageNet-1K | All models | `timm` Wightman (2019) |

## A.2 SYNTHETIC DATA SELECTION STRATEGY

Following the previous discussion on HOMO and HETERO, this section introduces our synthetic data selection strategy designed to mitigate the limitations of training discriminators on generative data.

The key difference from prior approaches is the incorporation of a diversity score. During selection, in addition to the fidelity score—which measures how closely synthetic samples resemble real data—the diversity score quantifies how far a synthetic instance departs from canonical patterns. As illustrated in Fig. 13, when referencing real instances, a synthetic sample receives a higher diversity score if its direction from the HETERO set deviates more strongly from the direction toward the canonical pattern.

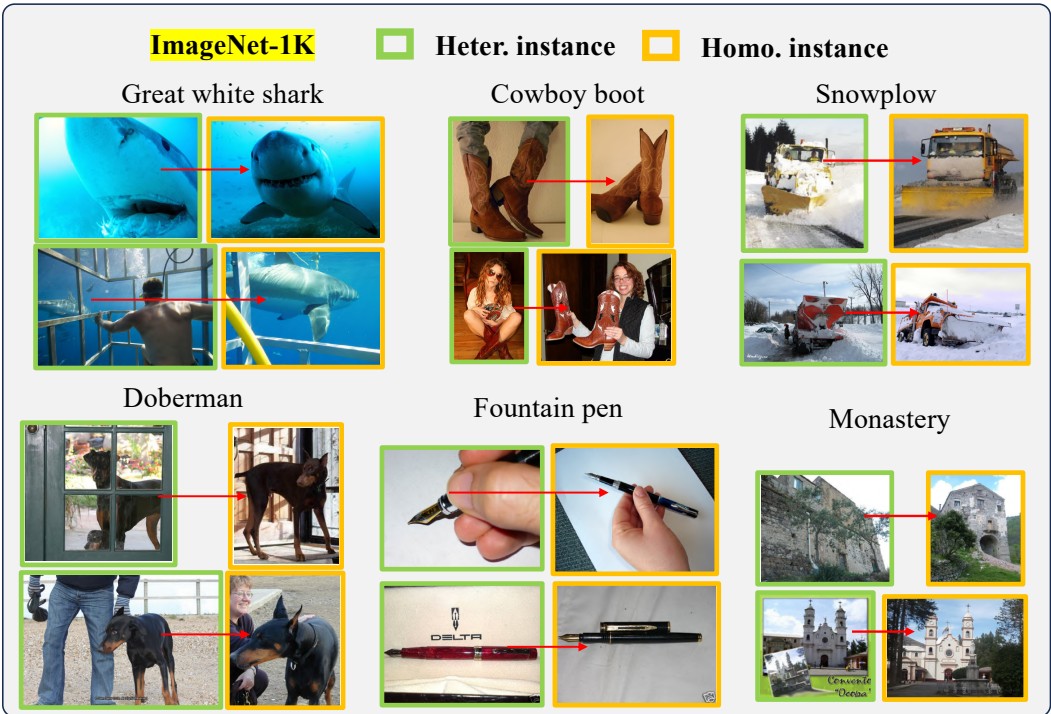

Figure 13: Examples of HETERO instances and their most similar counterparts in HOMO (based on MoCo V3 features) from ImageNet-1K. Red arrows denote vectors formed by each image pair. To capture the diversity within HETERO, our algorithm prioritizes synthetic images that, when paired with a **HETERO instance**, yield vectors that deviate from the red-arrow directions.

### A.3 EXPERIMENT DETAILS

To validate our strategy, we train models on synthetic datasets and evaluate them on real data across two settings: in-domain and out-of-domain testing. In these experiments, we utilize publicly available generative models and synthetic datasets, and conduct training using their released code repositories. The corrsponding resoures are illustrated in Tab. 5.

#### A.3.1 IN-DOMAIN CLASSIFICATION ACCURACY

We evaluate model performance on SVHN, CIFAR-10, and Tiny-ImageNet to enable hierarchical assessment across datasets of varying complexity, and further conduct experiments on ImageNet-1K to simulate the large-scale setting.

**SVHN:** We compare our method with prior approaches, and the results are reported in Tab. 6. When trained exclusively on synthetic datasets, model performance can surpass that of models trained on real data as the training volume increases. Among different strategies, ours consistently achieves the best performance across varying training data scales.

**CIFAR-10:** We compare our method with prior approaches, and the results are reported in Tab. 7. When trained exclusively on synthetic datasets, model performance can surpass that of models trained on real data as the training volume increases. Among different strategies, ours consistently achieves the best performance across varying training data scales.

Table 5: Public repositories and resources used in this work. The first three rows correspond to classifier training repositories, and the last two rows correspond to generative models.

| Resource | Link |
|---|---|
| **Classifiers** | |
| SVHN & CIFAR-10 | `https://github.com/ML-GSAI/Understanding-GDA` |
| Tiny-ImageNet | `https://github.com/DennisHanyuanXu/Tiny-ImageNet` |
| ImageNet-1K | `https://huggingface.co/docs/timm/training_script` |
| **Generative Models & Data** | |
| Synthetic data & code for SVHN, CIFAR, and Tiny-ImageNet | `https://github.com/wzekai99/DM-Improves-AT` |
| Generator for ImageNet-1K | `https://github.com/NVlabs/edm2` |

Table 6: **ResNet18** performance on the **SVHN** dataset across different training sizes. As the amount of synthetic training data increases, model performance improves. The **green bold text** indicates the highest accuracy across different data selection strategies.

| Method | 73,257 (Ori. Size) | 80K | 90K | 100K | 200K | 300K |
|---|---|---|---|---|---|---|
| *Real Data* | 95.84±0.12 | - | - | - | - | - |
| *RandSelect* (Wood et al., 2021) | 94.65±0.11 | 94.95±0.16 | 95.03±0.20 | 95.04±0.13 | 95.78±0.08 | 96.28±0.13 |
| *Clip-Align* (He et al., 2023) | 89.14±0.31 | 89.38±0.26 | 89.67±0.23 | 89.67±0.17 | 90.22±0.24 | 90.70±0.22 |
| *RealScore* (Kynkäänniemi et al., 2019) | 94.60±0.12 | 95.03±0.18 | 95.03±0.13 | 95.20±0.16 | 95.72±0.13 | 96.20±0.05 |
| *SBSim* (Lin et al., 2023) | 94.71±0.16 | 95.13±0.08 | 95.22±0.12 | 95.25±0.14 | 95.84±0.13 | 96.16±0.06 |
| *Ours* | **95.44±0.17** | **95.59±0.12** | **95.66±0.11** | **95.83±0.13** | **96.33±0.11** | **96.65±0.04** |

**Tiny-ImageNet:** We compare our method with prior approaches, and the results are reported in Tab. 8. In this more complex setting, we first train models exclusively on synthetic data, where the performance gap between synthetic- and real-trained models is around 10%. To effectively validate the utility of synthetic data in discriminator training, we use synthetic data as an augmentation to real data. Under this setting, our method consistently outperforms alternatives across different experimental configurations.

### A.3.2 Out-of-domain classification accuracy

To assess the robustness of trained models, we evaluate models on out-of-domain dataset.

**SVHN:** We evaluate OOD performance using the extra split of SVHN (Netzer et al., 2011) and test subset of distorted SVHN. Detailed results are presented in Tab. 9.

**CIFAR-10:** We use CIFAR-10-Warehouse (Sun et al., 2024) as a benchmark, the subset of data in this benchmark is illustrated in Fig. 14. The inference results are presented in Tab. 10.

**Tiny-ImageNet:** We utilize Tiny-ImageNet-C (Hendrycks & Dietterich, 2019a) for the evaluation of OOD, which incorporates various types of corruption. We classify them into three types: color-variation set (*i.e.,* brightness adjustment, contrast variation), noise-variation set (*i.e.,* pixelation, Gaussian noise, motion blur), and compression-variation set (*i.e.,* JPEG compression). The results are presented in Tab. 11.

**ImageNet-1K:** We use five ImageNet-1K OOD, illustrated in Fig. 15:

1) ImageNet-V2 (Recht et al., 2019) is constructed to closely match the distribution of the original ImageNet-1K, containing 50,000 images across the same 1,000 classes as the original validation set.

2) ImageNet-Sketch (Wang et al., 2019) consists of black-and-white sketches covering all ImageNet-1K classes, with 50 images per class.

Table 7: **ResNet18** performance on the **CIFAR-10** dataset across different training sizes. As the amount of synthetic training data increases, model performance improves. The **green bold text** indicates the highest accuracy across different data selection strategies.

| Methods | 50K (Original Size) | 70K | 90K | 100K | 200K | 300K | 400K | 500K |
|---|---|---|---|---|---|---|---|---|
| Real | 85.80 ± 0.73 | - | - | - | - | - | - | - |
| Random | 84.57 ± 0.31 | 87.74 ±0.48 | 88.73 ±0.71 | 89.57±0.25 | 92.78±0.18 | 93.94±0.25 | 94.68 ±0.08 | 95.03±0.06 |
| Clip | 77.09±1.19 | 80.97±1.05 | 82.34±0.27 | 82.81±0.67 | 85.62±0.64 | 86.94±0.14 | 87.56±0.29 | 88.39±0.25 |
| Realism | 83.95±1.28 | 87.45±0.58 | 89.03±0.39 | 89.70±0.16 | 92.60±0.30 | 93.92±0.44 | 94.64±0.20 | 94.93±0.11 |
| Similarity | 84.67±0.98 | 87.12±0.70 | 88.72±0.38 | 89.41±0.23 | 91.79±0.23 | 93.21±0.17 | 93.83±0.17 | 94.23±0.13 |
| **Ours** | **84.96±0.31** | **87.90±0.35** | **89.66±0.40** | **90.47±0.39** | **93.43±0.17** | **94.86±0.20** | **95.45±0.16** | **95.71±0.10** |

Table 8: Performance of EfficientNet-B0 and Resnet50 on the **Tiny-ImageNet** dataset with different synthetic dataset. Results are reported as Mean ± Std. (relative difference from the real-data baseline). The best results in each column are highlighted. As the amount of synthetic training data increases, model performance improves. The experiment results demonstrate the importance of the trade-off between fidelity and diversity. The original data size is 100K.

| | Data Selection Methods | +100K | +200K | +300K | +400K |
|---|---|---|---|---|---|
| EfficientNet-B0 | *Real Data* | 73.87 ± 0.29 | – | – | – |
| | *RandSelect* (Wood et al., 2021) | 74.16 ± 0.25 (+0.29) | 74.59 ± 0.28 (+0.29) | 75.00 ± 0.21 (+1.13) | 75.70 ± 0.14 (+1.70) |
| | *CLIP-Align* (He et al., 2023) | 60.54 ± 0.42 (−13.33) | 62.24 ± 0.50 (−11.63) | 62.59 ± 0.36 (−11.28) | 63.13 ± 0.05 (−10.74) |
| | *RealScore* (Kynkäänniemi et al., 2019) | 74.66 ± 0.27 (+0.79) | 75.67 ± 0.19 (+1.90) | 75.80 ± 0.14 (+1.93) | 76.60 ± 0.04 (+2.73) |
| | *MaxSim* ($\alpha = 0$) (Lin et al., 2023) | 74.69 ± 0.09 (+0.82) | 75.11 ± 0.25 (+1.24) | 75.48 ± 0.29 (+1.61) | 75.94 ± 0.16 (+2.07) |
| | *MaxDiv* ($\alpha = 1$) | 74.93 ± 0.11 (+1.06) | 75.59 ± 0.11 (+1.72) | 75.90 ± 0.11 (+2.03) | 76.23 ± 0.05 (+2.36) |
| | *Ours* | **75.03 ± 0.18** (+1.16) | **75.91 ± 0.06** (+2.04) | **76.65 ± 0.14** (+2.78) | **76.86 ± 0.14** (+2.99) |
| ResNet-50 | *Real Data* | 64.13 ± 0.67 | – | – | – |
| | *RandSelect* (Wood et al., 2021) | 66.55 ± 0.24 (+2.42) | 68.32 ± 0.25 (+4.19) | 69.45 ± 0.42 (+5.32) | 70.41 ± 0.21 (+6.28) |
| | *CLIP-Align* (He et al., 2023) | 53.12 ± 0.88 (−11.01) | 55.16 ± 0.38 (−8.97) | 56.97 ± 0.30 (−7.16) | 58.30 ± 0.25 (−5.83) |
| | *RealScore* (Kynkäänniemi et al., 2019) | 66.05 ± 0.45 (+1.92) | 69.12 ± 0.11 (+4.99) | 70.09 ± 0.19 (+5.96) | 71.51 ± 0.11 (+7.38) |
| | *SBSim* ($\alpha = 0$) (Lin et al., 2023) | 65.69 ± 0.35 (+1.56) | 67.12 ± 0.57 (+2.99) | 70.10 ± 0.37 (+5.97) | 71.26 ± 0.42 (+7.13B) |
| | *Diversity* ($\alpha = 1$) | 65.87 ± 0.50 (+1.74) | 68.16 ± 0.25 (+4.03) | 70.12 ± 0.13 (+5.99) | 72.52 ± 0.21 (+8.39) |
| | *Ours* | **68.57 ± 0.19** (+4.44) | **70.82 ± 0.08** (+6.69) | **71.80 ± 0.26** (+7.67) | **73.05 ± 0.25** (+8.92) |

3) ImageNet-C (Hendrycks & Dietterich, 2019b) evaluates model robustness to common corruptions such as noise, blur, weather effects, and digital distortions by applying perturbations with severity levels from 1 to 5 to the original ImageNet validation images.

4) ImageNet-Drawing (Salvador & Oberman, 2022) is derived from the ImageNet validation set, where images are transformed into drawing styles using generative adversarial networks and image processing techniques.

5) ImageNet-Cartoon (Salvador & Oberman, 2022) is also derived from the ImageNet validation set, where images are transformed into cartoon styles.

We categorize them into two main groups based on the sources of image. 1) Original OOD: ImageNet-V2 and ImageNet-Sketch; 2) Derivative OOD: ImageNet-C, ImageNet-Drawing and ImageNet-Cartoon. The inference results is illustrated in the Tab. 12, 13.

## A.4 ANALYSIS AND DISCUSSION

### A.4.1 LIMITATION OF CLIP FILTER

Reviewing the experimental results, we observe that using CLIP as a filter to select synthetic data consistently leads to the worst performance when training discriminators. Upon inspecting the selected samples, we attribute this degradation to the dominance of monotonous instances. As illustrated in Fig. 16, high- and low-CLIP-score examples in Tiny-ImageNet generated by EDM highlight this issue, while Fig. 17 presents corresponding cases in ImageNet-1K generated by EDM2. When constructing datasets from a synthetic pool, prioritizing high-CLIP-score instances produces collections enriched with canonical patterns but lacking diversity, ultimately resulting in weaker performance on both in-domain and out-of-domain evaluations.

## A.5 IMPACT OF DIFFERENT EXTRACTORS

Our strategy is built upon image feature representations; therefore, we further investigate whether the choice of feature extractor influences the final performance of classifiers. In this ablation study, we conduct experiments on two datasets: CIFAR-10 and ImageNet-100. The experimental con-

Table 9: **OOD** Performance comparison on **SVHN**. The best performance for each dataset is highlighted in **bold**.

| Method | SVHN | SVHN-Extra | SVHN-test-small-collage | SVHN-test-middle-collage | SVHN-test-strong-collage | SVHN-test w/large-collage | SVHN-Extra w/large-collage |
|---|---|---|---|---|---|---|---|
| Real Data | 95.90 | 98.09 | 93.53 | 86.24 | 80.20 | 72.15 | 75.68 |
| RandSelect | 96.19 | 98.20 | 93.85 | 86.52 | 80.92 | 73.35 | 77.27 |
| SBSim | 96.25 | 98.22 | 94.04 | 86.79 | 80.63 | 72.16 | 76.29 |
| Clip-Align | 87.21 | 92.77 | 83.97 | 73.26 | 63.83 | 54.37 | 60.87 |
| RealScore | 96.25 | 98.20 | 93.86 | 86.47 | 80.86 | 73.09 | 77.04 |
| **Ours** | **96.66** | **98.45** | **94.42** | **87.41** | **81.50** | **73.96** | **77.76** |

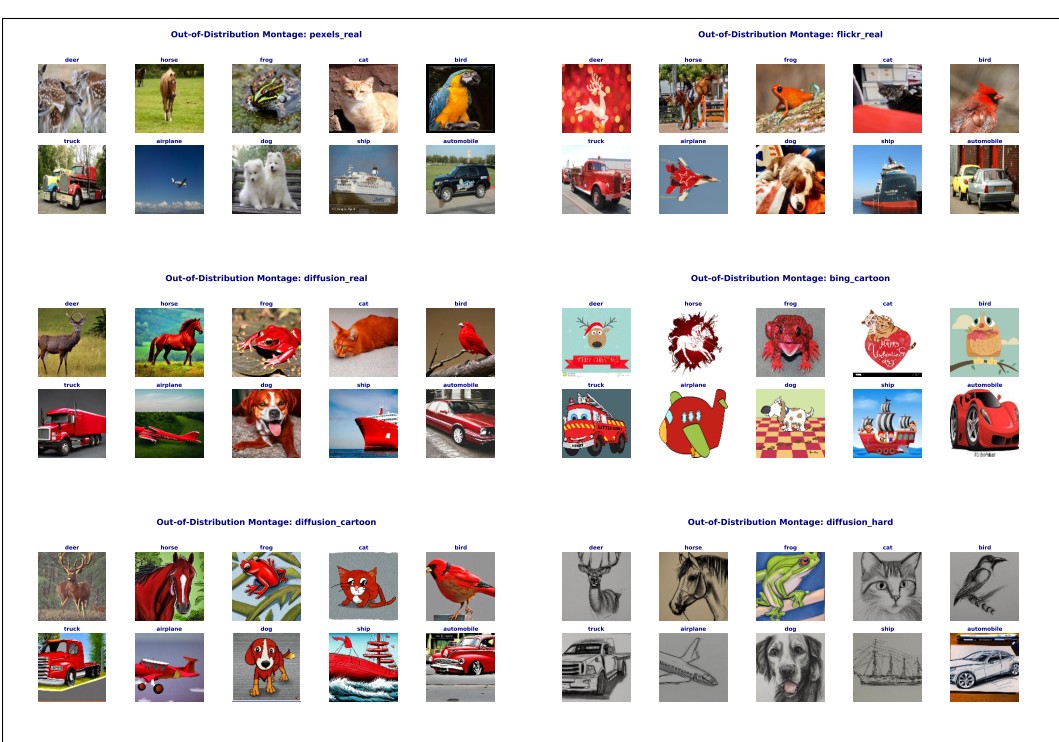

Figure 14: Subsets of CIFAR-10-Warehouse dataset for CIFAR-10 OOD testing.

figurations are summarized in Tab. 14. Specifically, for CIFAR-10, we use EDM to generate 1M synthetic samples and apply our strategy to select 100K training instances, which are then used to train a ResNet-18 model.

### A.5.1 IN-1K TRAINING RECIPE

We outline the training configuration adopted for training classifiers on the ImageNet-1K dataset. We train `vit_base_patch16_224` and `resnet50` from scratch, and fine-tune `vit_base_patch16_224.augreg_in21k` and `resnetv2_50x1_bit.goog_in21k` pretrained on ImageNet-21K.

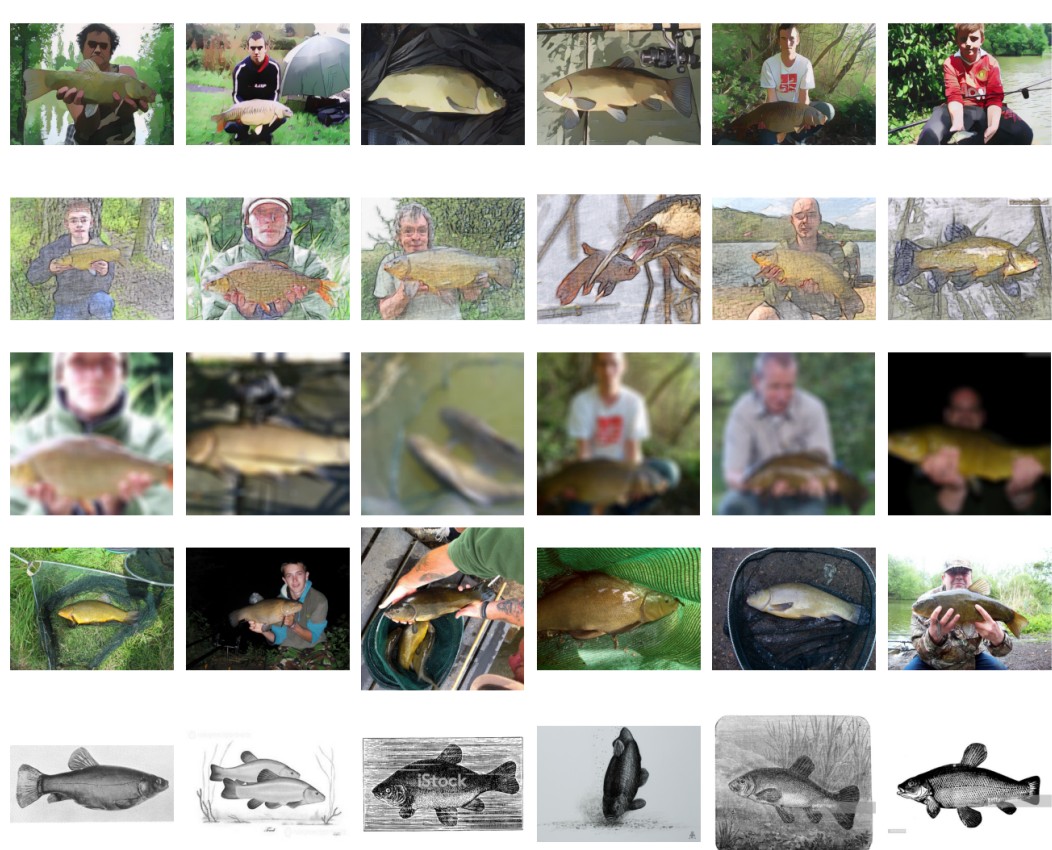

Figure 15: Subsets of IN-1K OOD data

Table 10: **OOD** Performance comparison on **CIFAR-10**. The best performance for each dataset is highlighted in **bold**.

| Method | Diffusion | Flickr | Pexels | 360 |
|---|---|---|---|---|
| Original | 0.819 | 0.708 | 0.810 | 0.547 |
| Random Sampling | 0.904 | 0.819 | 0.908 | 0.655 |
| Similar Only | 0.914 | 0.820 | 0.908 | 0.663 |
| CLIP | 0.932 | 0.797 | 0.898 | 0.660 |
| Realism | 0.924 | 0.831 | 0.920 | 0.679 |
| **Ours** | **0.934** | **0.850** | **0.935** | **0.709** |

Table 11: **OOD** Performance comparison on **Tiny-ImageNet**. The best performance for each dataset is highlighted in **bold**.

| Method | Color | Noise | Compression |
|---|---|---|---|
| Original | 40.52 | 46.94 | 48.33 |
| Random Sampling | 48.17 | 46.13 | 54.23 |
| Similar Only | 48.40 | 53.62 | 56.89 |
| CLIP | 40.52 | 39.73 | 40.12 |
| Realism | 49.41 | 52.57 | 54.23 |
| **Ours** | **52.19** | **58.77** | **61.86** |

Table 12: **OOD** Performance comparison on **ImageNet-1K** for ViT-B/16. The best performance for each dataset is highlighted in **bold**.

| Method | IN-V2-freq. | IN-Sketch | IN-Cartoon | IN-Drawing | IN-C (gaussian) | IN-C (motion) | IN-C (jpeg) | IN-C (snow) |
|---|---|---|---|---|---|---|---|---|
| Original | 49.86 | 11.213 | 43.36 | 17.494 | 18.434 | 30.774 | 47.668 | 29.476 |
| Random Sampling | 60.57 | 23.826 | 58.974 | 27.728 | 35.51 | 41.136 | 61.364 | 40.986 |
| SBSim | 60.16 | 23.3 | 58.554 | 29.342 | 33.258 | 39.736 | 61.158 | 39.952 |
| CLIP-Align | 56.3 | 21.193 | 56.276 | **32.686** | 35.086 | 38.16 | 58.296 | 38.08 |
| Realism | 60.61 | 23.73 | 58.93 | 27.75 | 35.48 | 41.2 | 61.31 | 40.88 |
| **Ours** | **61.25** | **25.308** | **59.65** | 30.138 | **37.226** | **43.394** | **62.18** | **42.666** |

Table 13: **OOD** Performance comparison on **ImageNet-1K** for ResNet50. The best performance for each dataset is highlighted in **bold**.

| Method | IN-V2-freq. | IN-Sketch | IN-Cartoon | IN-Drawing | IN-C (gaussian) | IN-C (motion) | IN-C (jpeg) | IN-C (snow) |
|---|---|---|---|---|---|---|---|---|
| Original | 56.98 | 17.34 | 41.852 | 14.948 | 11.418 | 22.78 | 41.552 | 22.46 |
| Random Sampling | 60.78 | 25.29 | 55.27 | 26.054 | 31.518 | **26.152** | 53.132 | 35.906 |
| SBSim | 58.81 | 24.378 | 52.118 | 24.762 | 28.18 | 25.774 | 50.156 | 30.628 |
| CLIP-Align | 56.45 | 24.201 | 49.996 | 24.956 | 27.188 | 22.928 | 49.212 | 27.722 |
| Realism | 60.78 | 25.3 | 55.25 | 26.044 | 31.6 | 26.13 | **53.135** | **35.906** |
| **Ours** | **61.76** | **26.77** | **55.404** | **26.16** | **35.458** | 25.93 | 52.98 | 34.29 |

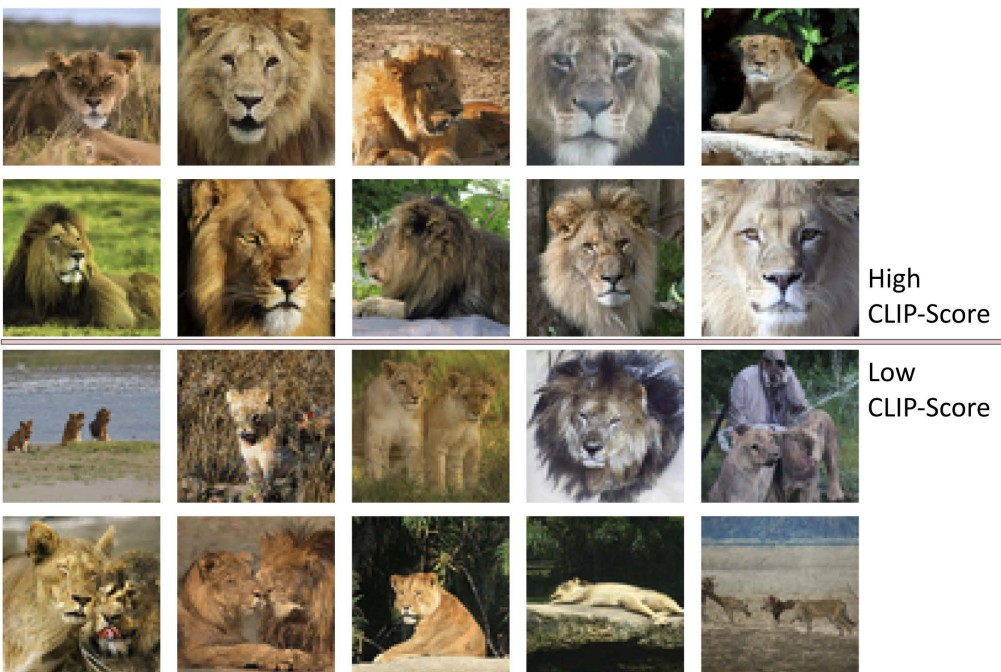

Figure 16: Instances with high and low CLIP-Scores in EDM generated Tiny-ImageNet.

Table 14: Experimental configurations for different datasets, including generator, classifier, and training dataset size.

| Dataset | Generator | Synthetic Data Size | Classifier | Training Size |
|---|---|---|---|---|
| CIFAR-10 | EDM | 1M | ResNet-18 | 100K |
| ImageNet-100 | EDM2 | 1M | ResNet-50 | 120K |

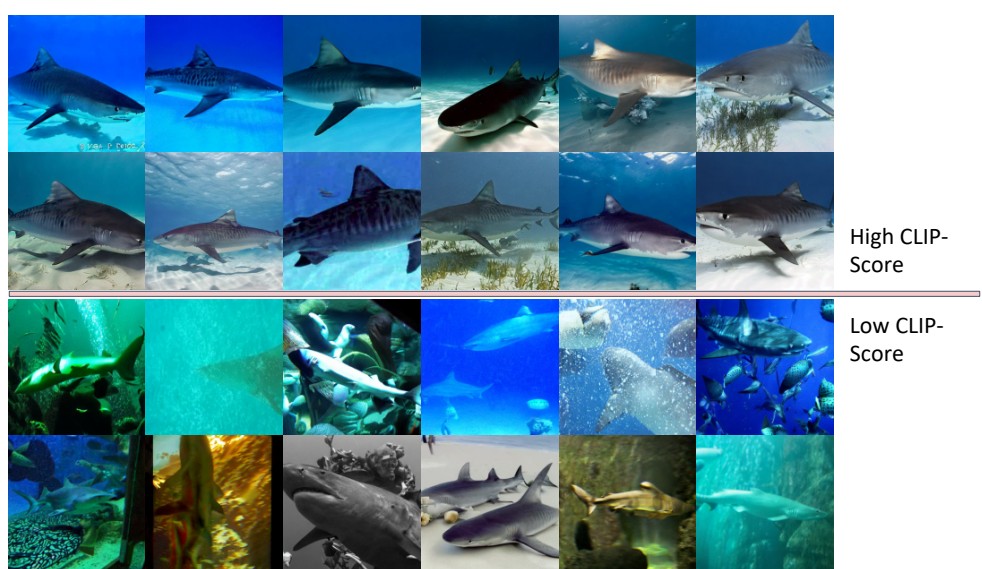

Figure 17: Instances with high and low CLIP-Scores in EDM2 generated ImageNet-1K.

```
1  ./distributed_train.sh "$NUM_GPU" \
2    --data-dir "$IMAGENET_DIR" \
3    --model vit_base_patch16_224 \
4    --epochs 120 --batch-size 256 \
5    --opt adamw --weight-decay 0.05 \
6    --sched cosine --warmup-epochs 5 \
7    --lr-base 5e-4 --lr-base-size 1024 \
8    --aa rand-m9-mstd0.5-inc1 \
9    --mixup 0.8 --cutmix 1.0 --smoothing 0.1 \
10   --drop-path 0.1 --channels-last --amp
```

Listing (1) ViT-B/16 training from scratch.

```
1  ./distributed_train.sh "$NUM_GPU" \
2    --data-dir "$IMAGENET_DIR" \
3    --model resnet50 \
4    --epochs 100 --batch-size 64 \
5    --opt sgd --momentum 0.9 --weight-decay 1e
        -4 \
6    --sched cosine --warmup-epochs 3 --
        cooldown-epochs 10 \
7    --lr 0.025 --smoothing 0.1 \
8    --mixup 0.2 --cutmix 1.0 \
9    --aa rand-m9-mstd0.5-inc1 \
10   --drop-path 0.05 --amp --channels-last
```

Listing (2) ResNet-50 training from scratch.

```
1  ./distributed_train.sh "$NUM_GPU" \
2    --data-dir "$IMAGENET_DIR" \
3    --model vit_base_patch16_224.augreg_in21k
        \
4    --pretrained \
5    --epochs 50 --batch-size 512 \
6    --opt adamw --weight-decay 0.05 \
7    --sched cosine --warmup-epochs 5 \
8    --lr-base 5e-5 --lr-base-size 1024 \
9    --aa rand-m9-mstd0.5-inc1 \
10   --mixup 0.2 --cutmix 0.8 --smoothing 0.1 \
11   --drop-path 0.1 --channels-last --amp
```

Listing (3) Fine-tuning ViT-B/16 (IN-21K init).

```
1  ./distributed_train.sh "$NUM_GPU" \
2    --data-dir "$IMAGENET_DIR" \
3    --model resnetv2_50x1_bit.goog_in21k \
4    --pretrained \
5    --epochs 50 --batch-size 64 \
6    --opt sgd --momentum 0.9 --weight-decay 1e
        -4 \
7    --sched cosine --warmup-epochs 3 --
        cooldown-epochs 10 \
8    --lr 0.025 --smoothing 0.1 \
9    --mixup 0.2 --cutmix 1.0 \
10   --aa rand-m9-mstd0.5-inc1 \
11   --drop-path 0.05 --amp --channels-last
```

Listing (4) Fine-tuning ResNet-50 (IN-21K init).

Figure 18: Training recipes for different backbones. Top row: from-scratch training. Bottom row: fine-tuning with ImageNet-21K initialization.

## A.6 CODE: SCORING

```python
def compute_synthetic_common_alignment_scores(
    syn_normed,                 # [N, D]   normalized synthetic features
    real_common_normed          # [M_1, D] normalized real-common):
    synthetic_common_fidelity = syn_normed @ real_common_normed.T

    # diversity score
    centroid_common = F.normalize(real_common_normed.mean(dim=0, keepdim=
    True), dim=1)
    A = syn_normed.unsqueeze(1) - real_common_normed.unsqueeze(0)  # [N,
    M_1, D]
    B = centroid_common - real_common_normed  # [M_1, D]
    numerator = (A * B.unsqueeze(0)).sum(dim=-1)      # [N, M_1]
    A_norm = A.norm(dim=-1)                            # [N, M_1]
    B_norm = B.norm(dim=-1).unsqueeze(0)               # [1, M_1]

    synthetic_common_diversity = numerator / (A_norm * B_norm + 1e-8)   #
    [N, M_1]
    synthetic_common_score = synthetic_common_fidelity -
    synthetic_common_diversity  # [N, M_1]
    return synthetic_common_fidelity, synthetic_common_diversity,
    synthetic_common_score

def compute_synthetic_rare_alignment_scores(
  syn_normed = None, real_rare_normed = None,
  real_np = None,  # real features matrix
  class_rare_2_common_in_real_dict = None  ):
  synthetic_rare_fidelity = syn_normed @ real_rare_normed.T  # [N, M_2]

  rare_2_common_in_real_dict = class_rare_2_common_in_real_dict

  common_counterpart_index = [i["common_image"] for i in
    rare_2_common_in_real_dict.values()]

  common_counterpart_matrix = get_real_subset_features_matrix(real_np,
    common_counterpart_index)  # [M_2, D]
  normalized_common_counterpart_matrix = F.normalize(
    common_counterpart_matrix, dim=1)  # [M_2, D]

  rare_2_common_matrix = normalized_common_counterpart_matrix -
    real_rare_normed  # shape: [M2, D]
  norm_rare_2_common_matrix = rare_2_common_matrix / rare_2_common_matrix
    .norm(dim=-1, keepdim=True)  # Normalize to unit vectors

  rare_2_synthetic_matrix = syn_normed.unsqueeze(1) - real_rare_normed.
    unsqueeze(0)  # [N, M2, D]
  norm_rare_2_synthetic_matrix = rare_2_synthetic_matrix /
    rare_2_synthetic_matrix.norm(dim=-1, keepdim=True)  # [N, M2, D]

  cos_sim_matrix = (norm_rare_2_synthetic_matrix *
    norm_rare_2_common_matrix.unsqueeze(0)).sum(dim=-1)
  synthetic_rare_diversity = cos_sim_matrix  # [N, M2]
  synthetic_rare_score = synthetic_rare_fidelity -
    synthetic_rare_diversity  # [N, M2]
  return synthetic_rare_fidelity, synthetic_rare_diversity,
    synthetic_rare_score
```

Listing 5: Synthetic alignment scoring

## A.7 CODE: SELECTION

```
def get_the_highest_score_index(matrix, syntheic_name_list, top_k=250,
    top_n=2, reverse=False):
    if matrix.shape[0] > matrix.shape[1]:
        matrix = matrix.T
        number_of_real = matrix.shape[0]
        number_of_synthetic = matrix.shape[1]

    if reverse:
        matrix = -matrix

    # for each row return the top-n largest column indices and values
    topk_values, topk_indices = torch.topk(matrix, top_n, dim=1)  # shape
    : (rows, top_n)
    # print("the shape of matrix:", matrix.shape)
    assert len(syntheic_name_list) == matrix.shape[1], \
        "Synthetic image names length must match the number of synthetic
    features."

    retrieval_syn_image_and_scores = {}

    # iterate through each row
    for indices, values in zip(topk_indices, topk_values):
        for idx, val in zip(indices, values):
            current_image_name = syntheic_name_list[idx.item()]
            current_value = retrieval_syn_image_and_scores.get(
    current_image_name, -9999)
            if val.item() > current_value:
                retrieval_syn_image_and_scores[current_image_name] = val.
    item()

    # check unique count
    assert len(set(retrieval_syn_image_and_scores.keys())) >= top_k, \
        f"The number of unique top-{top_n} column indices is less than
    top_k."

    return list(retrieval_syn_image_and_scores.keys()),
    retrieval_syn_image_and_scores
```

Listing 6: Top-k synthetic selection

## A.8 JUSTIFICATION OF HOMO-HETERO PARTITION

In the main paper, we first split the real data into two exclusive sets: HOMO and HETERO. In this section, we will further illustrate the motivation of such HOMO and HETERO design. Theoritically justify the optimization of our partition under paper setting.

### A.8.1 SEMANTIC-MODE DIVERSITY AND VARIATION DIVERSITY

To clarify the design of our real data partition, we first distinguish between semantic-mode diversity and variation diversity. We provide a PCA visualization in Fig.19. In Fig.19a, the real data are partitioned into two sets based on the distance of each instance to the class centroid: the "near" set contains canonical patterns, while the "far" set consists of patterns that are less similar to the semantic core and therefore exhibit higher semantic-mode diversity. Each of them cover the subregion of original feature space. In contrast, Fig. 19b illustrates our HOMO/HETERO partition, where both HOMO and HETERO cover all semantic modes present in the real data, and the diversity in HETERO manifests as variation diversity, rich variations around each semantic mode across the entire dataset.

Thus, Fig.19a illustrates semantic-mode diversity: the "near" and "far" sets are distinguished by the presence of different semantic modes in each subregion. In contrast, Fig.19b shows the

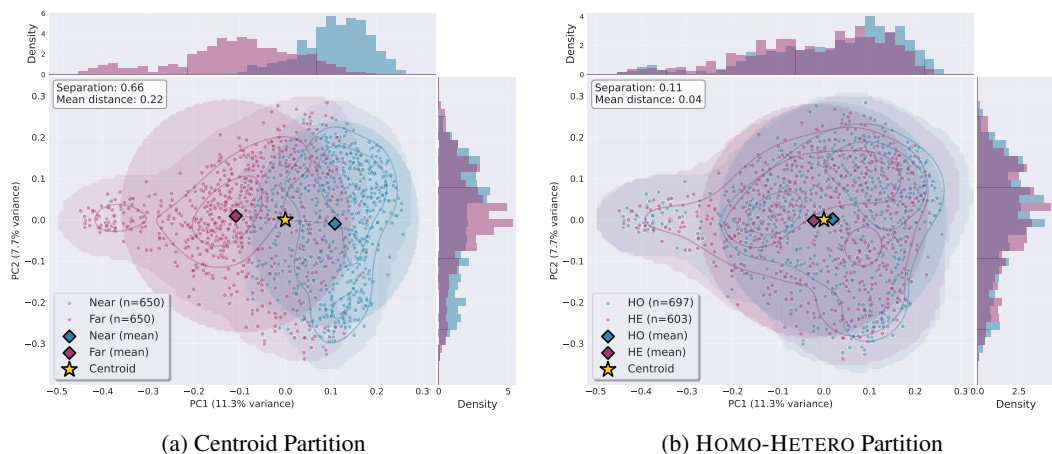

(a) Centroid Partition  (b) HOMO-HETERO Partition

Figure 19: PCA visualization in ImageNet-1K (n03769881), using different partition strategies

HOMO/HETERO partition, where the two sets are distinguished by the variation level within each mode across the entire feature space, rather than by different modes themselves.

### A.8.2  WHY WE CHOOSE HOMO-HETERO PARTITION

As illustrated in Sec. 3.1, standard image augmentations improve model performance by increasing variation within the dataset while preserving the main semantics. Inspired by this, we hypothesize that, when curating synthetic data, if we can construct a set that covers all semantic modes while exhibiting high variation within each mode, we can similarly expect performance gains. Based on this idea, we propose the HOMO/HETERO partition of the real data, which preserves all modes (as shown in Fig.19b) but explicitly distinguishes them by their level of intra-mode variation, and use them as guidance for synthetic data selection.

### A.8.3  THE CHARACTERISTIC OF HOMO-HETERO PARTITION

In this section, we discuss the properties of the HOMO –HETERO partition and prove its optimality under our setting.

As defined HOMO –HETERO in Eqs. 1–4,, we can review such partition in the graph setting. In the original dataset, each image feature could be viewed as the node, and the cosine similarity between different features could be used as the distance connecting each pair of nodes (higher similarity corresponds to a smaller distance). Then we can compose a fully connected graph $G = (V, E)$, and $V$ is the set of nodes, and $|V| = n$, and $E$ is the set of edges, $|E| = n \times n$.

To find the "hubs", we traverse all nodes in the graph and connect each node to its single nearest neighbor, forming a directed graph $G' = (V, E')$, where $|E'| = n$. Under this setting, HOMO is equivalent to the collection of nodes whose in-degree is positive.

So in the graph setting, the definition of HOMO could be rewritten as:

$$\text{Homo} = \{a \in V \mid \exists b \in V \setminus \{a\}, \text{ such that } \forall c \in V \setminus \{a,b\}, d(a,b) \leq d(c,b)\}$$

Based on the definition above, we can derive the following properties:

1. To reach any node $b$ in the graph, starting from a node in HOMO (different from $b$) consistently yields the minimal cost. Such property implies that learning the pattern in HOMO, and then we can reconstruct the whole original feature space with the smallest cost.

2. Suppose, in the original graph, each node has a single nearest neighbor. Then, theoretically, HOMO is a minimal subset of nodes such that every node in the graph can be reached from some node in HOMO with minimal cost.

