# OpenReview forum: "Balancing Fidelity and Diversity: Synthetic data could stand on the shoulder of the real in visual recognition"
_ICLR.cc/2026/Conference — ICLR 2026 Conference Withdrawn Submission_

### Official Review · Reviewer_ayvz · 2025-10-15

**Soundness:** 1
**Presentation:** 3
**Contribution:** 2
**Rating:** 2
**Confidence:** 5

**Summary:**

Authors look at the problem of synthetic training data, which they look at through the lens of dataset selection. In this way, they are able to examine the effects of balancing fidelity and diversity. The method itself consists of first dividing the real data into a homogenous set and a heterogeneous set. This is defined by first extracting features and comparing pairwise cosine similarities--any images that are the closest neighbor are considered homogenous, and the rest are heterogeneous. The synthetic data is split into these sets by scoring the cosine similarity with each of the sets. They evaluate on SVHN, CIFAR-10, Tiny ImageNet, and ImageNet, on ViT-B/16 and ResNet-50, and in both training-from-scratch and fine-tuning settings. They also evaluate OOD generalization, and perform additional analysis on impacts of the real data, diversity and fidelity trade-off, scaling, and feature extractor choice.

**Strengths:**

S1) Certain parameters (training setting = from scratch / fine-tuned, model = ViT-B/16 / ResNet-50, feature extractor = SigLIP / DINOv3 / ViT) are well explored.

S2) The fidelity-diversity trade-off is well-defined in literature, making it an interesting area to explore.

S3) The high-level motivations of the paper are written clearly.

S4) To my knowledge, this type of set-balancing has not been previously explored.

**Weaknesses:**

Major Weaknesses:

W1) The related works are missing important context, specifically in two broad areas:
W1a) This is very related to dataset sub-selection, as defined in [A]. It would be interesting to discuss how this work draws from that literature.
W1b) In the field of synthetic data, the choice of generation model is incredibly significant. It would be interesting to discuss the base models.

W2) In section 3.1, the experiment seems misaligned with the downstream task. Specifically, this section measures the trade-offs between fidelity and diversity, as a way to justify the methods later. However, here diversity is applied with low-level data augmentation techniques (cropping, rotation, color jittering). Diversity in this form is not what we may expect from a generative model, which may generate the object unrealistically or entirely wrong. Therefore, these seem to be the wrong augmentations to fit with the later work.

W3) This work proposes a very specific strategy for selecting the two sets in the real data (if an image is a nearest neighbor), but this choice is never ablated although there are other straightforward options (e.g. calculating a class centroid and classifying by distance). This makes it unclear if this is the best strategy. The contribution would be stronger if selection strategy were discussed or ablated.

W4) I find some claims in the paper too strong for the evidence provided. Specifically:
W4a) Line 216/217 claim: GMs more readily learn and reproduce HOMO
set containing repeated canonical pattern. However, it involves two unrelated datasets: the (generally fixed, when fixed to a specific generative model) synthetic distribution and the target real data distribution. Hence, whether a generative model generates HOMO or HETERO images depends on the target real distribution (perhaps in the clearest-to-imagine case, a target dataset could be constructed that is closer to HETERO). This statement would be much clearer if it were more specific, for example discussing natural datasets, the datasets in this work, distribution similarities between the test dataset and the training dataset for the generative model, etc, and it supported the claims.
W4b) In line 419, the 'dataset complexity' is referenced. This is a bit vague, and the statements would be stronger if it were defined more precisely.
W4c) The statements in the section in lines 448-452 seem strong for the evidence provided. Specifically, there are other unexplored factors that may impact the performance difference--e.g. this work uses MoCo V3, SigLIP, DINOv3, and ViT for features, but CLIP score uses CLIP.

W5) The HETERO partition seems to be computed very different mathematically for the synthetic data (distance from reference features) than the real dataset (not similar to any individual sample), which introduces some methodological inconsistencies. This choice would be stronger if there were strong discussion / justification for this choice.

W6) The datasets used do not provide strong enough validation. Specifically: a) there are too few when compared to other literature, b) some of them have much lower resolution than most generative models, and therefore results are not as convincing (e.g. Tiny ImageNet, CIFAR10), c) half of them have very low number of classes compared to many modern datasets (10), d) there is relatively little diversity between subjects. I say these specifically with respect to other literature in this area, which uses a much more diverse set of 10 datasets including broad / fine-grained classes, higher resolution, many datasets with 100+ classes. Two papers (also not cited in the related works) which use this are [B] and [C].

W7) The paper uses a full-dataset setting, but does not compare with other literature using this in other ways (e.g. full fine-tuning, just training with the real data instead of synthetic data, training with real and synthetic data as done in [B]). Comparing directly to these other methods in both resource consumption and performance is critical to fully understanding the best uses of synthetic data.

Minor Weaknesses:

w1) Most of the figures are not referenced in the text.

w2) The introductory paragraphs of the experiments seem to miss some important information that could be helpful for readers. Specifically, training setting (from scratch or real), how Sp is chosen, choice of generative model.

w3) The arrangement of figures seems a bit scattered--some (e.g. 6) are very far from their references, making them hard for readers to locate.

w4) The naming of the data subsets may be reconsidered--the words have other inappropriate connotations, and homo specifically is widely considered a slur.

[A] DataComp: In search of the next generation of multimodal datasets, Gadre et al., NeurIPS 2023.
[B] DataDream: Few-shot Guided Dataset Generation, Kim et al., ECCV 2025.
[C] Diversified in-domain synthesis with efficient fine-tuning for few-shot
classification, da Costa et al.

**Questions:**

Q1) I don't fully understand the scoring mechanism for the partitions. The equation for Sp in line 267 refers to cosine similarity between two sets, but how are these values aggregated? Also is this happening at a set level or an individual level for each image? (the Fs are defined as sets, but perhaps it is scoring each element in the synthetic set with the full real partition?)

Q2) I am also confused about how fidelity and diversity are scored--in line 257, it is said that it is repeated for each partition (HOMO, HETERO). However, then fidelity and diversity are both calculated from a partition p--does this mean there are 4 values? fidelity + diversity for HOMO, and fidelity + diversity for HETERO?

Q3) Which generative model is used? I did not find it in the text, and it is highly impactful.

---

> ### Author Response · Authors · 2025-11-22
> **Strength Summary and Responses to Weaknesses 1-3**
>
> We appreciate reviewer ayvz’s valuable effort and are glad about the positive feedback on our work specifically on: **comprehensive experiments**, the **interesting fidelity–diversity trade-off**, the **clear motivation and presentation**, and the **novelty of our set-balancing pipeline**.
>
> For the proposed weaknesses, we provide further justification.
>
> ## [W1] Missing important background
> (1) For reference papers, we agree on the benefit of a richer background and add these references in the revised paper ([A] in L58, [B] in L52, [C] in L129).
> (2) For generator usage, as stated in Sec. 4.1, we use EDM and EDM2 because they are well-established and provide checkpoints, simulating a setting with high-quality synthetic data. While other works rely on general-purpose generators like Stable Diffusion, whose outputs are weaker than domain-specific models and therefore require filtering of low-quality images. Our study, instead, focuses on curating already high-fidelity synthetic data, a more meaningful and underexplored direction as modern generative models continue to improve.
>
> ## [W2] Misalignment in Sec.3.1
> The experiments presented in the Preliminary Analysis are **intended to highlight the trade-off between data fidelity and diversity**, even when training in CIFAR-10.
> Since the diversity of generated data is difficult to control, as the reviewer noted: excessive diversity can introduce unrealistic or noisy samples, we conducted the analysis using real images. In this controllable setting, varying the diversity intensity does not alter the core semantics of the images, allowing for a more reliable examination of the fidelity–diversity balance than experiments based on highly diverse synthetic data (which may contain more noise).
> **In short, Sec.3.1 aims to first illustrate the impact of the fidelity–diversity trade-off on real datasets, and then motivate our proposed pipeline under the synthetic data setting. This design aims to facilitate the comprehension of readers, especially those without a strong background in this subfield**.
>
> ## [W3] Real data partition
> We understand the concern about the soundness of our real data partition, and here we will **first discuss the concept about diversity**, and then **theoretically justify the optimization of our real partition strategy**.
> **(1).** About diversity, using centroid could effectively separate dominated and rare modes within class: **(semantic) mode diversity**. With guidance of such partition, it **encourages synthetic data to grasp rare modes**. In our HO-HE partition, we emphasize **variation diversity** and prioritize the **retrieval of all modes with high variation**.
> **(2)**. We use HO-HE for two main reasons: 1. advanced generators can capture the full modes under targeted training, so missing rare modes is not the main issue of use the synthetics. 2. increasing the rare pattern ratio in curated dataset does not guarantee the robustness of downstream models, or may compromise the performance of dominated patterns.
>
> As shown in Sec. 3.1, image augmentation is a way to increase variation diversity while maintaining the (semantic) mode diversity, which lead to the effective improvement. Based on it, our HO-HE partition is optimal to capture the full modes in real data, and combining with our curation method, our pipeline could guarantee the retrieval of full modes while increasing the diversity of each concrete semantics.
> For why our HO-HE could capture full mode, we provide the theoretical proof (graph perspective) in the response of reviewer Z8Sg: [[W2] Missing theory grounding of Homo-Hetero partition](https://openreview.net/forum?id=6r0VuH8gGT&noteId=gD40a9JKJI).
> Duo to the space limit of rebuttal, we update the further justification about our HO-HE partition in Appendix A.8.
>
> **For empirically supporting**, we implement toy experiment in CIFAR10 setting, and guild the data curation with near/far partition:
> | Syn. Volume | 100k  | 200k  | 300k  | 400k  |
> |-------------|-------|-------|-------|-------|
> | near/far    | 89.56 | 92.86 | 94.05 | 94.61 |
> | ours        | **90.47** | **93.43** | **94.86** | **95.45** |
>
> These results directly support our claim: although the near/far split separates samples by distance to the class centroid, curation based on this heuristic does not guarantee better performance. Each subset covers only part of the original modes, and favoring either one does not necessarily improve performance over the full mode distribution.

---

> ### Author Response · Authors · 2025-11-22
> **Responses to Weaknesses 4-7 and Minor Weakness**
>
> ## [W4] Too strong claims
> We respond based on our understanding of the author's concerns:
> a) **Generated and real data are related**, since generators are training to mimic the real data distribution. In Fig. 4, our claim is that when generators are trained on the target real dataset, the resulting synthetic corpus more closely resembles the Ho subset of that dataset. If the omission of training-data details caused any misunderstanding, we apologize and will update the caption of Fig. 4 accordingly.
> b) **As we mention at the end of that paragraph, we use intra-class variation to reflect the complexity** of a dataset. When intra-class variation is low, the real data contains fewer diverse modes (as in SVHN), which naturally makes it harder to retrieve distinct synthetic data than in datasets with higher intra-class variation (like IN-1K).
> c) We are not entirely sure we fully understand this concern, but our interpretation is that **the weakness might stem from limitations of the CLIP encoder**. If so, **we believe this does not weaken our claim**: regardless of whether CLIP is strong or weak, our conclusions are drawn from the output of CLIP-score filtering (i.e., the subset it selects), not from assumptions about its internal mechanism. If this is not what you meant, could you kindly clarify your point in more detail?
>
> ## [W5] Inconsistency of HETERO in the synthetic and real
> **We do not select Hetero in synthetic data**.  Let’s review the whole curation pipeline:1.splitting real data into HO and HE, 2.use HO/HE to guild the synthetic data selection.
> Concretely, we score the fidelity of synthetic data based on the distance between the synthetic and real. For diversity, the HO and HE sets provide reference directions to measure the deviation from the canonical pattern. Combining fidelity and diversity, we get a final score for data selection.
>
> ## [W6] Not strong enough validation
> After reviewing [B, C], we would like to clarify **that these 2 works are parallel to ours**. [B, C] focus on adjusting the generative process in order to curate datasets, whereas our pipeline performs curation after a synthetic pool has already been generated.
> **Differ from them, our work focuses on analyzing the data distributions of real and generated samples**, and using this insight to refine an existing dataset, rather than intervening in the generation process itself.
>
> **Furthermore, we believe our experiments are sufficient to validate our idea**: the 4 datasets we use are all standard in the CV community and span a clear hierarchy of complexity (10, 200, and 1000 classes), with varying levels of difficulty and solution diversity, different data configurations (real+synthetic, synthetic only),  different training configurations (from scratch, fine-tunning).
>
> ## [W7] Missing multiple data configurations
> Thank you for the suggestion. However, we need to clarify that **we have already included multiple data configurations** in the paper.  In Fig. 6, the dashed line shows the performance of using only real data for clearer comparison. In Table 1, `+0` denotes training exclusively on real data, while `+xM` indicates training on the combination of real data and `xM` synthetic samples. In contrast, the xM entries in the last two columns correspond to training with `xM` synthetic samples only, without any real data. Similar setting in Tab. 8.
>
> ## Minor Weakness
> 1. Could you kindly point out which figure references are missing? We will promptly correct them. We apologize for any inconvenience caused.
> 2. **Beginning from L339**, we state the train data configuration while describing each experimental setting (synthetic-only, combination of real and synthetic ). For $S_p$ selection, as mentioned in **L289**, while training a model with K samples, we retrieve the top K unique sample with the highest score. For generative model, as mentioned in **L302, L305**, which choice EDM and EDM2 as they as established diffusion model optimized in target dataset and release their checkpoints.
> 3. We will reorganize the placement of the figures to achieve a more coherent and reader-friendly flow throughout the paper.
> 4. We apologize for any inappropriate connotations, we may refine the name in the revised version. In current version, I keep use such terminology avoiding any inconsistencies with discuss other reviewers.

---

> ### Author Response · Authors · 2025-11-22
> **Responses to Questions**
>
> ## [Q1] Fidelity calculation
> In Eq.5, Both $F^{syn}, F^p$ are matrices, and the this calculation return a cosine similarity matrix, recording the pairwise similarities.
>
> ## [Q2] Scoring cross HOMO and HETERO
> Each synthetic instance is evaluated against the entire real dataset, which is partitioned into HO and HE. Thus, synthetic dataset produces 2 scoring matrices: M1 of size $N_{Ho}$ × $N_{Syn}$ and M2 of size $N_{He}$ × $N_{Syn}$. For each synthetic–real pair, the corresponding matrix entry is computed as α * fidelity + (1 – α) * diversity.
> ($N_{x}$ indicates the number of samples in the dataset $x$)
>
>
> ## [Q3]  Generators Usage
> Regarding the generators, as stated in the first paragraph of Sec. 4.1 (Datasets and Baselines), we use the established EDM and EDM2 models in our main experiments. We chose these models because they are well-optimized diffusion models trained on the target datasets and provide publicly released checkpoints, ensuring full reproducibility.
>
> In the ablation study of α (see Fig. 9), we use StyleGAN (Karras et al., 2020), EDM (Karras et al., 2022), and EDM2 (Karras et al., 2024b) to analyze the impact of varying synthetic data quality. We also describe the generator usage for this ablation in the paragraph starting at L453.

---

> > ### Comment · Reviewer_ayvz · 2025-11-26
> >
> > I would like to thank the authors for their responses.
> >
> > W1, W3-5, W7, mW2-4, Q1-3. Thank you for your work, these answers are suitable to me and I now understand the clarifications.
> >
> > W2. I understand the purpose now, it is fine.
> >
> > W6. I understand your point, but I still hold my original view for the reasons discussed in the first review.
> >
> > mW1. This was my error--I had searched for them under "Figure" but they were referenced as "Fig.". They all seem to be there.
> >
> > While many of my concerns were addressed, the issue of the impact of the experimental section remains and heavily impacts my rating (see W6). For now, I will keep my original score.

---

> ### Author Response · Authors · 2025-11-28
>
> We are glad to hear that our previous responses solve the majority concerns and question.
>
> For `W6`, we appreciate the reviewer’s thoughtful comments regarding dataset coverage and diversity. Below we clarify our design choices and how our evaluation aligns with, and in some aspects exceeds, existing works (e.g., [B], [C] as suggested by the reviewer).
>
> >1. Number of datasets
>
> While we train our method on 4 in-domain datasets, we evaluate in a zero-shot and OOD setting in a total of **13 datasets**, including multiple domain shifts. This results in broader evaluation coverage than the 2 few-shot generative works cited by the reviewer ([B], [C]), which typically evaluate on around 10 datasets but focus primarily on in-domain classification. We apologize that this was not sufficiently emphasized in Sec. 4.1, as most of these datasets were described later in the OOD evaluation section (Sec. 4.3). We have clarified this in the revised version (**L306**).
>
> >2. Resolution and number of classes
>
> (1) Our method is **a selection mechanism**, not an image generator; therefore, image resolution is determined by the underlying datasets (e.g., ImageNet at 224×224).
> (2) We intentionally use datasets with **a wide range of resolutions and number of classes** to demonstrate generalizability. We demonstrate that our method works well in all types of dataset from simple to complex: SVHN and CIFAR-10 (small-scale, low-resolution benchmarks), Tiny-ImageNet (mid-scale with 200 classes), and ImageNet-1K (large-scale with 1000 classes).
> (3) In many real-world domains such as surveillance, remote sensing, and medical imaging from low-cost equipment, low-resolution datasets are very common. Demonstrating strong performance across resolutions is thus important for real-world applicability.
> >3. Dataset diversity
>
> Our evaluation covers **4 diverse in-domain datasets and 9 OOD datasets spanning various domains, scales, and resolutions**.
> We also note that ImageNet-1K (already included) is the most widely accepted benchmark in the CV community, covering 1000 classes and offering substantial category diversity. Many of the reviewer-suggested datasets (Caltech101, Pets, Food-101, Flowers-102) are either smaller or substantially overlap with ImageNet categories. Others focus on specific niche domains (e.g., textures (DTD); remote sensing (EuroSAT); fine-grained categories (Aircraft, Cars); scene classification (SUN397)). We agree these datasets are valuable, but their domain specificity makes them less directly relevant for evaluating general-purpose synthetic data selection, especially when our OOD evaluation already includes similarly diverse domains.
> >4. Dataset comparison with prior works
>
> To provide transparency, we include a comparative table in the supplementary material summarizing all datasets used in our paper versus those used in the cited works [B] and [C]. This highlights that our evaluation is comparable or broader in both scale and domain coverage.
> Paper|Datasets Used|Domain Coverage|Resolution Range|#Classes Range/Avg
> -|-|-|-|-
> Ours|13 datasets: CIFAR-10, SVHN, Tiny-ImageNet, ImageNet-1K, Tiny-ImageNet-C, CIFAR-10-Warehouse, ImageNet-V2, ImageNet-Sketch, ImageNet-C, ImageNet-Drawing, ImageNet-Cartoon, SVHN-extra, Distorted-SVHN | Generic objects, digits, corruption robustness, distribution-shift robustness, sketches, cartoons|32×32→*224×224*|10→1000/496.2
> [B] & [C]|10 datasets: ImageNet, Oxford Pets, FGVC Aircraft, Food-101, Stanford Cars, DTD, EuroSAT, Flowers-102, SUN397, Caltech-101|Generic objects, fine-grained categories, scene, satellite images|64×64→*224×224* |10→1000/209.1
>
> (We use *224×224* here as the resolution setting to downsample higher-resolution inputs during training; the same configuration is also used in [B, C], according to their released code.)
> >5. Generation model
>
> Our scope focuses on **selecting high-quality synthetic data after generation**, and we choose such 4 datasets since they are widely used benchmarks in modern high-fidelity diffusion studies. Established models (e.g., EDM, EDM2) trained on these datasets generate high-fidelity data, fitting our setting.
>
> In contrast, Caltech101, Pets, Food-101, Flowers-102, DTD, EuroSAT, Aircraft, Cars, Pets are less used in the mainstream high-fidelity diffusion training, and lack widely adopted, high-quality open-source diffusion checkpoints or released synthetic datasets. Although some prior works use general-purpose generators like GLIDE or Stable Diffusion to synthesize clones of these datasets, the resulting images are typically less well aligned with the targeted distributions. In this low-fidelity regime, optimizing the generators toward the target distributions is crucial for improving synthetic data utility (, which is precisely what [B, C] aim to do).
>
> Therefore, to realistically simulate the use of high-quality synthetic data, we employ these four well-studied datasets and their derivatives for evaluation.

---

> > ### Author Response · Authors · 2025-11-28
> > **All citations used in the previous post**
> >
> > Here is the all citations used in previous [post](https://openreview.net/forum?id=6r0VuH8gGT&noteId=SniYRiaxpA)
> >
> > [B] DataDream: Few-shot Guided Dataset Generation, Kim et al., ECCV 2025.
> > [C] Diversified in-domain synthesis with efficient fine-tuning for few-shot classification, da Costa et al.
> > [D] IS SYNTHETIC DATA FROM GENERATIVE MODELS READY FOR IMAGE RECOGNITION? ICLR 2023.
> > [E] Scaling Laws of Synthetic Images for Model Training ... for Now. CVPR2024.

---

### Official Review · Reviewer_JRWP · 2025-10-25

**Soundness:** 1
**Presentation:** 1
**Contribution:** 2
**Rating:** 2
**Confidence:** 5

**Summary:**

This paper investigates how balancing fidelity and diversity in synthetic data selection impacts model performance. The study employs both a standard image classification task and an out-of-distribution classification task to analyze this relationship. Fidelity and diversity are quantitatively defined by comparing each synthetic image with its corresponding real reference image. Based on this analysis, the paper proposes a novel synthetic data selection method, and experimental results show that applying this method improves accuracy compared to existing selection approaches under various setups, including training from scratch, fine-tuning, and using different dataset scales (1M and 3M synthetic samples).

**Strengths:**

- Exploring the relationship between fidelity and diversity addresses a very interesting, practical, and challenging problem in data generation.

- The approach of splitting the real dataset into HOMO and HETERO subsets to measure these two properties is impressive; the splitting method appears both novel and conceptually sound.

- Experimental results demonstrate that the proposed selection method effectively improves performance, confirming its practical utility across different training setups.

**Weaknesses:**

**Definition of Fidelity and Diversity**

a. The definition of fidelity and diversity appears conceptually unclear.

b. Algorithm 1 is difficult to follow:

- In line 1, does axis = 1 refer to the first axis (1-indexed) or the second axis (0-indexed)?

- In line 2, is a single sample, $\mathcal{F}^{ref}$, representing the entire synthetic dataset, or one per synthetic instance?

- Does it (lines 3–9) apply to each synthetic sample individually, or to the entire synthetic dataset at once?

- In line 4, $\mathcal{F}^{syn} \in \mathbb{R} ^ {s \times d}$ and $\mathcal{F}^{p} \in \mathbb{R} ^ {(n \text{ or } m) \times d}$ have different dimensions—how can cosine similarity be computed between them? A similar dimensionality issue arises in line 6 among $\mathcal{R}^{ref} \in \mathbb{R} ^ {d}$, $\mathcal{F}^{p} \in \mathbb{R} ^ {(n \text{ or } m) \times d}$, and $\mathcal{F}^{syn} \in \mathbb{R} ^ {s \times d}$.

c. Beyond these ambiguities, it is questionable whether a single sample (i.e., $\mathcal{F}^{ref}$) can represent the entire distribution of the HETER set.

d. Similarly, comparing each synthetic image only to one representative sample (either $\mathcal{F}^{ref}$ or $\mathcal{C}^{Homo}$) may not capture the full distributional diversity within the real subsets (HOMO and HETER).

**Presentation and Clarity of Experiments**

a. The experimental setup and results presentation require clarification.

- What is the main distinction between the classical classification and OOD tasks in the context of applying the proposed selection method? Are they treated simply as two independent tasks?

- In Table 1, the terms Random and Realism are undefined—please clarify what these baseline methods represent.

- What does “+0” mean in Random selection in Table 1? Does it indicate that no synthetic data were used?

- More details are needed on how synthetic images were generated from each dataset. For instance, if SVHN was used, was the generator trained on the SVHN training set and then used to arbitrarily generate samples?

- In Figure 8, how was the violin-plot distribution obtained? Were multiple synthetic datasets generated and evaluated?

**Minor Concerns**

a. Figure numbering is inconsistent, and some figures are placed far from their first reference, making the paper difficult to follow.

b. Although the proposed selection method shows slightly better accuracy than compared methods in Table 1, the margin is small. A discussion comparing computational cost versus accuracy improvement would be beneficial to assess the method’s practical value.

c. There is some confusion between the main manuscript and the supplementary materials. Several figures that are substantially discussed in the main paper are located in the supplementary section. Reviewers should not be required to read the supplementary materials to fully understand the main content.

**Questions:**

- In Figure 9, why does balancing between fidelity and diversity by controlling $\alpha$ affect the generators differently?

**Details Of Ethics Concerns:**

I don't have any ethical concern.

---

> ### Author Response · Authors · 2025-11-22
> **Strength Summary and Responses to Weaknesses**
>
> We appreciate the effort of reviewer JRWP and are glad to hear the positive feedback on **our fidelity–diversity trade-off topic**, the **novelty and conceptual soundness of the HOMO–HETERO split**, and **the supportive experimental results**.
>
> For the proposed weaknesses, we provide further justification.
>
> # Definition of Fidelity and Diversity
> ## [W1] Unclear definition of fidelity and diversity
> Thank you for pointing this out. We clarify both terms below.
> **Fidelity** measures how well each synthetic sample aligns with real data. We quantify this using cosine similarity: in Eq. 5, the cosine similarity matrix records the “fidelity” of each synthetic sample relative to the real ones. Higher similarity indicates higher fidelity.
> **Diversity** measures how much a synthetic sample deviates from canonical patterns in the data. We quantify this using the angle-based metric: in Eq. 6, we compute the angle between the difference vectors composed by three features. Fig. 5 is draw for better illustration of deviation between two vectors(in red lines).
> We will clarify these definitions in the revised manuscript and are happy to incorporate additional reviewer suggestions.
>
> ## [W2] Clarity of Algorithm 1
> Thank you for the comment. We have clarified the algorithm in the text and added a Python implementation in Appendix A.6 & A.7. The main points are as follows:
> 1) In the algorithm (line 1), we follow PyTorch’s convention: `idx` is the set of indices of the maximum value in each row. Thus, axis=1 means taking the argmax over columns for every row.
> 2) In line 2, $F^{ref}$ is the **set of Ho features** closest to instances in He.
> 3) The for-loop in line 3-9 for the clarification of algorithm ideas and it is implemented in the **dataset level rather than the individual instance**. In programming, such operations are implemented by the matrix operation, which effectively get the score of whole synthetic set.
> 4) Line 4, 6 use pseudo-code to illustrate the core operations, not the exact low-level implementation. The code follows a python API (as below), which hides matrix details and exposes a simple matrix interface to highlight the idea rather than implementation details.
> ```python
> from sklearn.metrics.pairwise import cosine_similarity
> import numpy as np
> input1 = np.random.randn(3, 5)
> input2 = np.random.randn(3, 5)
> sim_matrix = cosine_similarity(input1, input2)
> ```
>
> ## [W3] Clarification of $F^{ref}$
> **$F^{ref}$, indeed, is a  collection of the features in Ho** which are most similar to the features in He. Different He instances can get different nearest features in Ho, so |$F^{ref}$|>1.
>
> ## [W4] Full distributional diversity
> **$F^{ref}$ contains multiple real features in Ho**. While calculating diversity in He, each real feature in He indeed get the direction to its most similar Ho features. **Since different He instances can have different nearest neighbors in Ho, they induce deviation vectors in diverse directions**. Therefore, our diversity metric gets an instance-level guidance which covers the full real data distribution.
>
> # Presentation and Clarity of Experiments
> ## [W5] Distinction between the In- and Out-of-domain classification
> In image classification, our evaluation follows two main sub-tasks: (1) in-domain classification and (2) robustness under distribution shift. These two aspects are commonly used to comprehensively assess data curation pipelines in prior work such as [1]. Therefore, we adopt a similar setting to validate our method.
> [1] DataComp: In search of the next generation of multimodal datasets. NeurIPS 2023.
>
> ## [W6] Issue in Tab.1
> *Random and Realism are undefined:*
> In the Tab.1, the Random and Realism are 2 baseline methods we compared, which is described in the Sec 4.1 baseline block.
>
> *unclear of `+0`*:
> Yes. As we mentioned in Sec 4.2, ImageNet-1K block, for setting 1) we use synthetic data as the augmentation of real data. +0 means there is no extra synthetic data used for training. Such columns are shown as a real data baseline for comparison.
>
> ## [W7] Unclearness of data generation
> As we mention in Sec. 4.1, Datasets block, we use generators which are trained in each targeted dataset. Tab. 5 in Appendix provides the concreted resources of them, and the dataset usage or generation process follows the guidelines in the corresponding official repositories.
>
> ## [W8] Unclearness of Fig.8
> We will refine the figure and caption for better comprehension. The process is:
> 1. generate a huge synthetic pool; 2. extract features with different encoders; 3. apply our curation pipeline; 4. train the same model with the same configuration multiple times. 5. get the multiple evaluations and draw the violin plot.
>
> In the original plot, each gray dot is one performance of one model after training.

---

> ### Author Response · Authors · 2025-11-22
> **Responses to Minor Concerns and Question**
>
> # Minor Concerns:
> ## 1. Figure placement and organization
> Thank you for the suggestion about figure placement, we have updated the pdf file for the better organization.
>
> ## 2. Discussion in computational cost versus accuracy
> We appreciate the reviewer’s point.
> **(1).** For the computational benefits of our curation, we report quantitative evaluation of computational cost (i.e., runtime) and model performance below:
> |||Random|||Ours||
> -|-|-|-|-|-|-
> Data size|Cur./Tr. Time|Total Time|Acc|Cur./Tr. Time|Total Time|Acc
> 200K|0/80|80|74.16|3.8/80|83.8|75.03
> 300K|0/120|120|74.59|3.8/120|123.8|75.91
> 500K|0/210|210|75.70|3.8/210|213.8|76.86
>
> (Cur./Tr. Time=Data Curation /Training Duration; Time in minutes, accuracy in %)
>
> In this setting, we use **MocoV3** to extract feature from **1M synthetic images**, and then train **EfficientNet_B0**, all of experiment run in **one NVIDIA L40S**. In such given setting, feature extraction take 3.5 minutes, Score & selection operation takes 0.3 minutes, thus, the whole curation takes extra 3.8 minutes besides training duration.
>
> Regardless of training volume, and this small extra cost saves substantial training time to reach better performance: Training 200K samples with our pipeline takes 83.8 minutes (accuracy 75.03),  outperforming 300K samples with random selection (120 minutes, accuracy 74.59), **saving ≈30% training time** while improving accuracy. Similarly, training 300K samples with our pipeline takes 123.8 minutes, outperforming 500K samples with random selection (210 minutes), **saving ≈41% training time**.
>
> **Since curation is a **one-time cost** for a fixed synthetic pool, whereas training cost grows with the number of samples and epochs, the relative overhead of curation decreases as training scale increases**. *Consequently, this efficiency gain of our pipeline becomes even more significant for larger synthetic training scales*.
>
> **(2).** The performance gains on ImageNet-1K in Table 1 are indeed modest, but this is expected and does not diminish the validity of our approach for several reasons:
> (i) ImageNet-1K is one of the most challenging and comprehensive datasets commonly used in the CV field. And the consistently outperformance of our in multiple settings in tab1 also provide strong evidence of the validation of ours.
> (ii) Another reason for the small performance gap in “Scratch” in Tab.1 is the impact of real data(1.2M) which takes more than 50% and 25% in +1M and +3M settings. The large volume of real data mitigate the difference of impact of different synthetic dataset.  While training exclusively in synthetic dataset in fine-tuning settings, the performance gap is more obvious. Thus while increasing the ratio of different synthetic datasets, the difference of synthetic gap will be more obvious in the large scale.
>
> ## 3. Confusion between the main manuscript and the supplementary materials
> We have revised the paper to improve clarity and ensure that the main content can be more easily followed without requiring content in appendix. Some figures referenced in the main paper were placed in the supplementary section due to page limit constraints. These figures are intended to complement the tables and figures in the main text, and all core arguments in the main paper are self-contained without requiring the supplementary material.
>
>
> # Question
> ## 1. Comprehension of Fig. 9
> Firstly, we want to clarify that the correspond subsection (*Impact of Generators and Data Scales on the Optimal Fidelity-Diversity Balance (α)*) of Fig. 9 focus on how different generators and data scales both impact the optimal hyperparameter $α$, rather than the opposite. As defined in Eq. 7, $α$ is used for scoring after the data generation. **Thus $α$ can not impact generators**. In Fig.9, we explore how different generators and training data scale impact the $α$, and provide our conclusion based on the experimental result in the figure.

---

### Official Review · Reviewer_aJnE · 2025-10-29

**Soundness:** 2
**Presentation:** 3
**Contribution:** 2
**Rating:** 4
**Confidence:** 3

**Summary:**

This paper addresses the challenge of selecting high-quality synthetic data for training visual recognition models. The authors propose a post-generation curation strategy that balances fidelity and diversity. The key innovation is partitioning real data into HOMO and HETERO subsets, then scoring synthetic samples based on their relationship to both partitions. Experiments across CIFAR-10, Tiny-ImageNet, ImageNet-1K, and SVHN demonstrate that this approach improves both in-domain accuracy and out-of-domain robustness compared to baseline selection methods.

**Strengths:**

1. Well-written
- The paper is clearly structured and easy to follow

2. HOMO-HETERO partitioning
- The intuitive split of real data based on intra-class similarity provides an interpretable framework for understanding data characteristics.

3. Comprehensive experiments
- The paper includes extensive experiments across multiple datasets (SVHN, CIFAR-10, Tiny-ImageNet, ImageNet-1K), architectures (ResNet, EfficientNet, ViT), and evaluation settings (in-domain and OOD).

4. Extensive analysis
- The paper provides insightful analysis on the relationship between synthetic data quality, fidelity-diversity trade-offs, and the impact of different feature extractors.

**Weaknesses:**

1. Limited theoretical justification
- While the HOMO-HETERO split is intuitive, the paper lacks theoretical grounding for why this particular partitioning strategy is optimal. The definition (Equations 3,4) is somewhat arbitrary.

2. Novelty
- The fidelity score is essentially cosine similarity, and the diversity score is a relatively straightforward angle-based metric. The main contribution is the partitioning strategy rather than the scoring itself. I hope the authors can discuss how to strengthen the novelty of this work.

3. Computational cost
- The paper doesn't analyze the computational overhead of feature extraction, similarity computation, which could be significant for large-scale datasets.

4. Dependence on feature extractor
- While Section A.5 provides some ablation, the choice of feature extractor (MoCo v3) has not been explored.

**Questions:**

1. The paper consistently uses $\alpha = 0.5$ throughout experiments, but Figure 9 shows that optimal α varies by dataset. Can you explain the rationale for fixing $\alpha = 0.5$?

2. As mentioned in the Weakness section, I would appreciate if the authors could discuss how to enhance the novelty of this contribution.

3. All experiments are on image classification. Can this work be expanded to other tasks to demonstrate generalization?

**Details Of Ethics Concerns:**

Because this paper proposes methods for synthetic data curation and selection, adding an Ethics section would be valuable.

---

> ### Author Response · Authors · 2025-11-22
> **Strength Summary and Responses to Weaknesses 1 & 2, and Question 2**
>
> We appreciate the valuable comments of aJnE, and are grateful for the positive feedback about the **clarity of our writing, the intuitive HOMO–HETERO partitioning, the comprehensive experiments, and the extensive analysis**.
>
> For the weaknesses, we provide further justification.
> ## [W1] Theoretical justification of Homo-Hetero partition.
> Thank you for the suggetions!
> In Sec. 3.1, image augmentation improves performance by preserving the original semantics while increasing per-instance variance. Motivated by this, HO–HE partition is proposed, aiming to identify 2 subsets that both capture the full semantics but differ in their level of variance. Then guilding the synthetic data curation.
> Based on this, our strategy is the optimal, which could be explained in the graph setting: features could be viewed as nodes, the similarity matrix (Eq. (1)) becomes the adjacency matrix of a fully connected graph $G=(V,E)$ where $|V|=n$, $|E|=n \times n$. The reverse of similarity serve as the transformation cost between features.
>
> Identifying hubs using a nearest neighbor graph $G'=(V,E')$, where $|E'|=n$. $G'$ is constructed by connecting each node to its single nearest neighbor with a directed edge. Homo is the set of nodes in $G'$ with in-degree>0.
>
> Thus, Homo is the **smallest collection of nodes**, which could **reach $\forall b\in V$ in minimal cost**. So:
>
> $Ho\in\arg\min\limits_{S\subseteq V}\sum_{b\in V}\min\limits_{\substack{a\in S\\ a\neq b}}e(a,b)$
>
> **This property of the Homo theoretically supports claims in L216, 222**: for both generators and classifiers, learning Homo is more favorable than Hetero, as Homo preserves the minimal cost between the learned and real holistic distributions.
>
> Since $He=V\setminus Ho$, Hetero also covers all modes while exhibiting higher variation. Thus, we use the Homo–Hetero partition in the following pipeline.
>
> Thank you again for the constructive suggestions. **We have added further theoretical details and illustrations in Appendix A.8 of the revised paper.**
>
> ## [W2 & Q2] Novelty of contribution
> We would like to clarify that, although cosine similarity and angle-based metrics themselves are not novel, our contribution lies in **how we apply them**.
> We restate our contribution and novelty in two aspects:
> 1. The Ho–He partition strategy, which reveals the preference of generator or classifier train on the data.
> 2. Based on the Ho–He, we proposed a scoring based curation method, which considers both fidelity and diversity while data selection.
>
> Regarding **real data partition**, our Homo-set selection provides a theoretical explanation of why such subsets are preferred by models after training (see W2). This offers new insight into how data composition shapes model behavior after training.
> Regarding **post-generation curation**, to our knowledge, our work is the **first explicitly introduce diversity metrics during curation**. Diversity becomes increasingly important as modern generators achieve ever higher fidelity. Previous strategies like K-means or SBSim retrieve high-fidelity samples but overlook less frequent yet semantically correct and valuable ones. Diversity metrics in post-generation data selection are still underexplored, and our proposed metric fill this gap. By jointly consider both fidelity and diversity of synthetic instances with respect to real data. our pipeline yields superior performance (Figs.6,7,9, Tab. 1), directly **shows the contribution of our works, especially as modern synthetic data already achieve high fidelity.**

---

> ### Author Response · Authors · 2025-11-22
> **Responses to Weaknesses 3 & 4**
>
> ## [W3] Computational cost
> Data curation (preprocessing) is vital for model training. Unlike unscalable manual curation, feature-based, model-driven curation scales well in modern pipelines. **Our method is one such approach, with modest and practically tolerable computational cost**.
>
> (1)**Theoretical upper bound**. Our curation pipeline has 2 parts: (A) feature extraction and (B) scoring & selection.
> Step A runs a modern feature extractor (e.g., CLIP) once over the dataset, which is much cheaper than training a modern model.
> Step B consists only of basic matrix operations (see Appendix A.6–A.7 for details) and costs far less than Step A.
> Thus, for the whole synthetic dataset, **the total curation cost is bounded by about $2f$, where $f$ is the cost of a single pass of the extractor over the dataset**.
>
> (2) **Empirical evidence**. We also report quantitative evaluation of computational cost (i.e., runtime). In the sbumitted paper, the Tiny-IN setting spans multiple data scales, enabling a direct and convincing efficiency comparison. In Tab.8 of the paper, our method outperforms random selection baseline even using less training images. The duration of model training and model performance shown as below:
>
> |||Random|||Ours||
> -|-|-|-|-|-|-
> Data size|Cur./Tr. Time|Total Time|Acc|Cur./Tr. Time|Total Time|Acc
> 200K|0/80|80|74.16|3.8/80|83.8|75.03
> 300K|0/120|120|74.59|3.8/120|123.8|75.91
> 500K|0/210|210|75.70|3.8/210|213.8|76.86
>
> (Cur./Tr. Time=Data Curation /Training Duration; Time in minutes, accuracy in %)
>
> In this setting, we use **MocoV3** to extract feature from **1M synthetic images**, and then train **EfficientNet_B0**, all of experiment run in **one NVIDIA L40S**. In such given setting, feature extraction takes 3.5 minutes, Score & selection operation takes 0.3 minutes, thus, the whole curation takes extra 3.8 minutes besides training duration.
>
> Regardless of training volume, and this small extra cost saves substantial training time to reach better performance:
>
> Training 200K samples with our pipeline takes 83.8 minutes (accuracy 75.03),  outperforming 300K samples with random selection (120 minutes, accuracy 74.59), **saving ≈30% training time** while improving accuracy.
> Similarly, training 300K samples with our pipeline takes 123.8 minutes, outperforming 500K samples with random selection (210 minutes), **saving ≈41% training time**.
>
> **Since curation is a **one-time cost** for a fixed synthetic pool, whereas training cost grows with the number of samples and epochs, the relative overhead of curation decreases as training scale increases**. *Consequently, this efficiency gain of our pipeline becomes even more significant for larger synthetic training scales*.
>
> ## [W4] Feature extractor matters
> We acknowledge that different encoders can produce substantially different absolute feature vectors for the same instances. **(1) However, the inter-instance relationships (which images are similar or dissimilar) tend to be much more stable across encoders**.
> To verify that different encoders perceive image relationships similarly, we performed **Representational Similarity Analysis (RSA)** on 10 random ImageNet classes. RSA compares pairwise similarity structure between models: 0 means two encoders disagree completely; while 1 means they see the same relational structure.
> ||SigLIP|DINO|MoCo
> -|-|-|-
> SigLIP|1|0.623|0.625
> DINO||1|0.762
> MoCo|||1
>
> The high off-diagonal RSA (0.62–0.76) confirm that, despite architectural and training differences, these encoders yield *very similar pairwise similarity structures* over the same set of images. In other words, the relational structure our method relies on is notably stable.
>
> **(2) Furthermore, instead of relying on a single absolute feature, our pipeline use all instances to vote for a representative subset, focusing on global semantics**. Although different encoders emphasize different visual aspects, the core semantics of a class remain stable under this holistic voting scheme.
> We further demonstrate **the selection outputs using our method exhibit strong cross-encoders overlap**. We analyzed whether different encoders lead to the selection of the same "HO-sets" (data subsets). Using CLIP as a baseline, we compared the overlap with sets chosen by SigLIP, DINO, and MoCo.
> ||SigLIP|DINO|MoCo|
> |-|-|-|-|
> |CLIP|0.63|0.62|0.65|
>
> These >60% overlaps show that the method consistently identifies the same core data points, regardless of the encoder used. The outcome is not tied to any single feature space.
>
> **(3) Empirically, as shown in Fig. 8 of the paper**, ablations across multiple encoders (*i.e.,* SigLIP, DINO-v3, ViT) reveal the choice of encoder has only a minor effect on final classification results. Across repeated runs, the framework’s behavior remains statistically stable, underscoring its robustness.
>
> In summary, our proposed pipeline remains stable and generalizable across diverse encoders and modalities.

---

> ### Author Response · Authors · 2025-11-22
> **Responses to Questions 1 & 3 and Ethics Concerns**
>
> ## [Q1]  Rationale for fixing α=0.5
> In main experiment, we set **$α=0.5$ to highlight the effect of balancing fidelity and diversity**, emphasizing the missing diversity factors in the prior curation.
> We acknowledge that $α$ impacts the final performance, so we conduct an ablation study in **Fig. 9 to provide a global picture of such hyperparameter** under various generators and training budgets.
>
> As a practical guideline, starting with $α=0.5$ is a reasonable choice to see the benefit of balancing fidelity and diversity. Once the fidelity and diversity matrices are computed, the final score is just their weighted sum, making it easy to search for a better $α$ (e.g., via binary search) if one wants to find the optimum for a specific setting.
>
> ## [Q3] Expanding to other tasks
> We appreciate the reviewer's concern regarding the scope of evaluation. Our experiments focus on classification with two evaluation criteria: (1) in-domain accuracy and (2) robustness, following the standard evaluation protocol established in works like DataComp [1].
> We acknowledge that extending our method to segmentation, detection, and cross-modal tasks represents valuable future work. We will revise the manuscript to explicitly discuss these potential extensions and their expected benefits.
> [1] Li, Jeffrey, et al. DataComp-lm: In search of the next generation of multimodal datasets. NeurIPS 2024.
>
> ## Ethics concerns for synthetic data curation and selection
> We appreciate the reviewer’s suggestion. Because our paper proposes methods for synthetic data curation and selection, we agree that an Ethics section is appropriate. We have added a dedicated Ethics Statement (see Sec. 7 of the revised paper), discussing bias, environmental impact, privacy considerations, and responsible use in synthetic data curation. We believe this addresses the reviewer’s concern and strengthens the paper.

---

### Official Review · Reviewer_Z8Sg · 2025-11-01

**Soundness:** 2
**Presentation:** 2
**Contribution:** 2
**Rating:** 4
**Confidence:** 3

**Summary:**

This paper introduces a post-generation curation framework for selecting high-quality synthetic data to improve visual recognition models. The authors argue that both fidelity (semantic similarity to real data) and diversity (novel variations) are crucial for maximizing the utility of synthetic datasets. They propose partitioning real data into homogeneous (HOMO) and heterogeneous (HETERO) subsets and scoring synthetic samples by their fidelity and diversity relative to each subset. The method, which is generator-agnostic and training-free, consistently enhances in-domain and out-of-domain accuracy across various datasets and models. Experiments demonstrate that carefully balancing fidelity and diversity yields better generalization and robustness than existing data selection strategies.

**Strengths:**

1. The paper addresses a timely and practically important problem (i.e., how to effectively select synthetic data rather than merely generating it) and provides a simple yet principled solution.

2. The paper is well-written, well-organized, and provides clear visualizations that make the proposed method and its effects intuitively understandable.

**Weaknesses:**

1. The proposed framework relies heavily on the quality and representation power of the feature extractor (e.g., CLIP, SigLIP). As a result, the selection outcome may be biased or unstable when different embedding spaces are used, limiting the generality of the method across various encoders or modalities.

2. While the HOMO–HETERO partition is conceptually intuitive, it lacks a formal theoretical justification or analysis explaining why this separation leads to optimal or stable performance. Without such grounding, the effectiveness of the split may vary significantly across datasets with different intrinsic structures.

3. Although the authors claim that the proposed selection method is efficient and scalable, these aspects are discussed only qualitatively. The paper would benefit from a quantitative evaluation of computational cost, such as runtime or memory usage as a function of dataset size, to substantiate the scalability claim.

4. The proposed method appears to incorporate a core-set–like selection mechanism into the fidelity–diversity framework, aiming to choose a representative yet diverse subset of synthetic samples. While this idea is conceptually related to existing core-set selection principles, the paper does not clearly differentiate its approach or justify how it fundamentally extends beyond standard core-set algorithms in either theoretical formulation or empirical advantage.

**Questions:**

Please address the comments in weakness.

---

> ### Author Response · Authors · 2025-11-22
> **Strength Summary and Responses to Weaknesses 1 & 2**
>
> We appreciate the thoughtful comments of Z8Sg and are grateful for **positive feedbacks on our topic and proposed solution**, as well as the recognition of the **organization and visualizations in clarify our pipeline**.
>
> For the weaknesses, we provide further justification.
> ## [W1] Robustness to feature extractor choices.
> We need to clarify that our method relies on **how images relate to one another**, rather than the precise coordinates of individual features. Hence, the method is **highly robust** to the choice of feature extractor. Although different encoders produce embeddings in different spaces, the *relative* structure (*i.e.,* which images are similar or dissimilar) tends to be highly consistent. We demonstrate this robustness from three complementary perspectives.
> **1. Inter-instance relationships are consistent across encoders.** To verify that different encoders perceive image relationships similarly, we performed **Representational Similarity Analysis (RSA)** on 10 random ImageNet classes. RSA compares pairwise similarity sturcture between models: 0 means two encoders disagree completely; while 1 means they see the same relational structure.
> ||SigLIP|DINO|MoCo|
> |-|-|-|-|
> |SigLIP|1|0.623|0.625|
> |DINO||1|0.762|
> |MoCo|||1|
>
> The high off-diagonal RSA (0.62–0.76) confirm that, despite architectural and training differences, these encoders yield *very similar pairwise similarity structures* over the same set of images. In other words, the relational structure our method relies on is notably stable.
>
> **2. We further demonstrate the selection outputs using our method exhibit strong cross-encoders overlap**. We analyzed whether different encoders lead to the selection of the same "HO-sets" (data subsets). Using CLIP as a baseline, we compared the overlap with sets chosen by SigLIP, DINO, and MoCo.
> ||SigLIP|DINO|MoCo
> -|-|-|-
> CLIP|0.63|0.62|0.65
>
> These >60% overlaps show that the method consistently identifies the same core data points, regardless of the encoder used. The outcome is not tied to any single feature space.
>
> (3) **Empirically, as shown in Fig. 8 of the paper**, ablations across multiple encoders (*i.e.,* SigLIP, DINO-v3, ViT) reveal the choice of encoder has only a minor effect on final classification results. Across repeated runs, the framework’s behavior remains statistically stable, underscoring its robustness.
>
> In summary, while encoders differ in their raw feature spaces, the **relational structure among images is preserved**. Since our method operates on this underlying structure, it remains stable and generalizable across diverse encoders and modalities.
> ## [W2] Missing theory grounding of Homo-Hetero partition.
> Thank you for the suggetions! To provide theoretical grounding, we reinterpret the similarity matrix (Eq. (1) of our paper) as the adjacency matrix of a fully connected graph $G=(V,E)$, where $|V|=n$, $|E|=n\times n$. The inverse similarity defines the edge cost $e(a,b)$, representing the transformation cost between feature nodes.
>
> We then detect hubs via a nearest-neighbor graph $G'=(V,E')$, where each node connects to its single nearest neighbor through a directed edge. Nodes with in-degree greater than zero form the *Homo-set*.
>
> This construction yields a key property: **the Homo-set is the smallest subset of nodes that minimizes the total cost required to reach all nodes in the graph:**
>
> $Ho\in\arg\min\limits_{S\subseteq V}\sum_{b\in V}\min\limits_{\substack{a\in S\\ a\neq b}}e(a,b).$
>
> This minimal-cost covering property provides theoretical support for our claims (L216, L222). Because Homo-nodes form the lowest-cost representatives of the underlying data manifold, learning them first is more stable and reduces the training objective for both generators and classifiers. They capture the central, high-density modes of the real distribution with minimal distortion.
>
> The Hetero set, defined as $He=V∖Ho$, still spans all data modes but with greater variability. This complementary structure motivates the Homo–Hetero partition used throughout our pipeline.
>
> Thank you again for the constructive suggestions. **We have added further theoretical details and illustrations in Appendix A.8 of the revised paper.**

---

> ### Author Response · Authors · 2025-11-22
> **Strength Summary and Responses to Weaknesses 3 & 4**
>
> ## [W3] Quantitative analysis of our method's efficiency and scalability
> Data curation (preprocessing) is vital for model training. Unlike unscalable manual curation, feature-based, model-driven curation scales well in modern pipelines. **Our method is one such approach, with modest and practically tolerable computational cost**.
>
> (1)**Theoretical upper bound**. Our curation pipeline has 2 parts: (A) feature extraction and (B) scoring & selection.
> Step A runs a modern feature extractor (e.g., CLIP) once over the dataset, which is much cheaper than training a modern model.
> Step B consists only of basic matrix operations (see Appendix A.6–A.7 for details) and costs far less than Step A.
> Thus, for the whole synthetic dataset, **the total curation cost is bounded by about $2f$, where $f$ is the cost of a single pass of the extractor over the dataset**.
>
> (2) **Empirical evidence**. We also report quantitative evaluation of computational cost (i.e., runtime). In the sbumitted paper, the Tiny-IN setting spans multiple data scales, enabling a direct and convincing efficiency comparison. In Tab.8 of the paper, our method outperforms random selection baseline even using less training images. The duration of model training and model performance shown as below:
> |||Random|||Ours||
> -|-|-|-|-|-|-
> Data size|Cur./Tr. Time|Total Time|Acc|Cur./Tr. Time|Total Time|Acc
> 200K|0/80|80|74.16|3.8/80|83.8|75.03
> 300K|0/120|120|74.59|3.8/120|123.8|75.91
> 500K|0/210|210|75.70|3.8/210|213.8|76.86
>
> (Cur./Tr. Time=Data Curation /Training Duration; Time in minutes, accuracy in %)
>
> In this seting, we use **MocoV3** to extract feature from **1M synthetic images**, and then train **EfficientNet_B0**, all of experiment run in **one NVIDIA L40S**. In such given seting, feature extraction take 3.5 minutes, Score & selection opertion takes 0.3 minutes, thus, the whole curation takes extra 3.8 minutes besides training duration.
>
> Regardless of training volume, and this small extra cost saves substantial training time to reach better performance:
>
> Training 200K samples with our pipeline takes 83.8 minutes (accuracy 75.03),  outperforming 300K samples with random selection (120 minutes, accuracy 74.59), **saving ≈30% training time** while improving accuracy.
> Similarly, training 300K samples with our pipeline takes 123.8 minutes, outperforming 500K samples with random selection (210 minutes), **saving ≈41% training time**.
>
> **Since curation is a **one-time cost** for a fixed synthetic pool, whereas training cost grows with the number of samples and epochs, the relative overhead of curation decreases as training scale increases**. *Consequently, this efficiency gain of our pipeline becomes even more significant for larger synthetic training scales*.
>
> ## [W4] Differences to existing core-set selection principles
> Our distinctions lie in 2 core aspects:
> 1.**Core-set composing**. We propose the Ho–He partition, which novelly splits the real dataset into two subsets that preserve the full semantics of the original set but differ in variation.
> 2.**How to use the core-set**. We introduce a novel angle-based diversity metric and a fidelity–diversity balancing mechanism to more comprehensively assess and curate synthetic data.
>
> **Regarding core-set selection**, as justified in W2, the Homo set is a small subset that can reach the any nodes in graph with minimal cost. Methods like K-means requires manually tune K, ensuring core-sets capture whole mode with fine-grained diversity.
> Unlike the prior, our partition is fully data-driven and theoretically guarantees that both subsets cover all modes while differing in variation.(we have update the A.8 for detailed discussion about our Ho-He selection).
> **Regarding post-generation curation**, prior strategies such as k-NN, SBSim, or Realism retrieve high-fidelity samples by measuring closeness to core-sets, but often overlook rare yet semantically correct and valuable ones. **Leveraging the Ho–He partition, we propose a novel angle-based diversity metric and jointly balance it with fidelity**; our curation strategy then achieves superior performance (Figs. 6 and 7), directly highlighting the value of our Ho–He–based diversity design.

---

### Note · Authors · 2026-01-27

I have read and agree with the venue's withdrawal policy on behalf of myself and my co-authors.

---

### Meta-Review · Area_Chair_z3vC · 2026-01-04

**Summary:**

This paper receives ratings of 4, 4, 2, 2. Main concerns are 1) lack of theoretical justification; 2) limited generality across encoders and modalities; 3) lack of computational cost analysis; 4) insufficient experiments; 5) unclear details and typos. Although authors' rebuttal have addressed 1), 3) and 5), AC agrees with reviewers on 2) and 4). Experiments are only conducted on datasets with limited number of classes and low resolutions, especially considering the era of large models. Authors' responses didn't fully address this issue. Thus the decision is reject.

**Reviewer Concerns:**

Concerns on lack of theoretical justification, lack of computational cost analysis, unclear details and typos are addressed. While concerns on the comprehensiveness of experiments still remains.

**Reviewer Scores:**

Not certain about this as their concerns are addressed only partially.

---

### Decision · Program_Chairs · 2026-01-26

Reject